# State of the Art in Development of Heat Exchanger Geometry Optimization and Different Storage Bed Designs of a Metal Hydride Reactor

**DOI:** 10.3390/ma16134891

**Published:** 2023-07-07

**Authors:** Viktor Kudiiarov, Roman Elman, Natalia Pushilina, Nikita Kurdyumov

**Affiliations:** Division for Experimental Physics, School of Nuclear Science & Engineering, National Research Tomsk Polytechnic University, 634050 Tomsk, Russia; kudiyarov@tpu.ru (V.K.); pushilina@tpu.ru (N.P.); nek6@tpu.ru (N.K.)

**Keywords:** hydrogen energy, metal hydride, metal hydride reactor, heat exchanger, heat transfer, thermal management

## Abstract

The efficient operation of a metal hydride reactor depends on the hydrogen sorption and desorption reaction rate. In this regard, special attention is paid to heat management solutions when designing metal hydride hydrogen storage systems. One of the effective solutions for improving the heat and mass transfer effect in metal hydride beds is the use of heat exchangers. The design of modern cylindrical-shaped reactors makes it possible to optimize the number of heat exchange elements, design of fins and cooling tubes, filter arrangement and geometrical distribution of metal hydride bed elements. Thus, the development of a metal hydride reactor design with optimal weight and size characteristics, taking into account the efficiency of heat transfer and metal hydride bed design, is the relevant task. This paper discusses the influence of different configurations of heat exchangers and metal hydride bed for modern solid-state hydrogen storage systems. The main advantages and disadvantages of various configurations are considered in terms of heat transfer as well as weight and size characteristics. A comparative analysis of the heat exchangers, fins and other solutions efficiency has been performed, which makes it possible to summarize and facilitate the choice of the reactor configuration in the future.

## 1. Introduction

Hydrogen is a promising alternative energy carrier. Over the last century, hydrogen energy has been used in aviation and aerospace, the defense industry and in fuel cell vehicles. In this context, hydrogen has become a promising option due to its high calorific value and lack of negative environmental impact. However, since a hydrogen energy system is a large and complex engineering system that includes the production, storage and use of hydrogen, the implementation of such a system is associated with a number of difficulties. One of the key challenges for large-scale hydrogen applications is the implementation of safe and efficient storage and transportation of hydrogen. However, an important problem limiting the use of hydrogen is hydrogen storage [1,2]. There are three aggregate states in which hydrogen is stored: gaseous, liquefied and adsorbed/absorbed. Hydrogen gas is mainly stored as a compressed gas, which is the optimal short-term option for hydrogen-powered vehicles. At the moment, hydrogen storage vessels can be classified into four types: Type I, which is an all-metal storage tank; Type II, which is a liner hoop-wrapped composite vessel; Type III, which is a fully wrapped composite cylinder with a metal liner that acts as the hydrogen permeation barrier; and Type IV, which is a fully wrapped composite cylinder with a plastic or composite liner. Type I vessels are inexpensive solutions, but they are heaviest and have low safety. Type IV storage tanks are the best possible solution for storing hydrogen at 70 MPa with good safety and minimal losses and leakage due to the extensive use of composite materials to ensure hydrogen impermeability through the cylinder wall. However, storage tanks operating at this pressure have low hydrogen density and are extremely expensive due to the need for improved composite materials [3,4]. The hydrogen density and hydrogen content can be increased by liquefying. Liquefaction makes it possible to store hydrogen with a density of 70 kg/m^3^ at atmospheric pressure. However, the storage of hydrogen in liquid form requires an energy-intensive liquefaction process by precooling and maintaining a temperature of −253 °C using liquid nitrogen. In addition, the process of storing hydrogen in liquid form is usually accompanied by high losses of hydrogen [5,6]. Chemical methods are promising means of hydrogen storage. A huge number of materials and compounds have been studied. The main advantage of storing hydrogen in chemical compounds such as ammonia, methanol or other cycloalkanes is the high volumetric density. A significant advantage is that chemical hydrides are usually liquids under normal conditions, which simplifies their transportation and storage. The main difficulty in storing hydrogen in this state lies in the limitations or lack of reuse of the material and the need for special plants for processing and obtaining pure hydrogen [7]. Porous adsorbent materials are considered as materials for hydrogen storage by physisorption due to their high surface-area-to-volume ratio and the ability of hydrogen to be adsorbed on the inner and outer surface walls. Examples of porous materials for hydrogen storage are carbon nanomaterials, aerogels, zeolites, and metal–organic frameworks (MOFs). This method provides safely and energy-capacitive hydrogen storage [8,9,10]. Hydrogen storage in metal hydrides is the one of the safest methods as well [11,12]. Metal hydrides exhibit acceptable hydrogen storage properties at low pressures, and their bulk density is comparable to that of liquid hydrogen. Such significant advantages of metal hydrides have led them to become increasingly important as hydrogen storage materials at the present time. Alloys based on rare earth metals, titanium, zirconium, magnesium and other hydride-forming metals are the basis for new hydrogen storage materials. Their use makes it possible to store, absorb and desorb large amounts of hydrogen. Depending on the specific application, a certain metal hydride can be selected. Thus, LaNi_5_ is able to absorb about 1.4 wt.% hydrogen at room temperature with the formation of LaNi_5_H_6_ hydride, while magnesium is able to absorb up to 7.6 wt.% hydrogen at a temperature of about 400 °C [13,14,15]. Nevertheless, the storage of hydrogen in hydride-forming metals is accompanied with drawbacks, the solution of which has received considerable attention from researchers. Thus, over the past decade, a large number of catalytic alloying additives have been identified that can improve the behavior of metal hydrides during hydrogen sorption and desorption reactions [16,17,18,19].

However, solid state hydrogen storage and purification systems have serious disadvantages related to thermal effects during the hydrogenation/dehydrogenation process. These effects significantly affect metal hydride reactor performance. Therefore, the implementation of efficient heat transfer in the metal hydride bed can significantly improve the characteristics of the metal hydride reactor. Known methods for increasing heat transfer include the use of foam metals, compacts (compressed hydride-forming material admixed with high thermal conductivity materials), heat exchangers, and a phase change materials.

Despite the large number of solutions to improve heat transfer in a metal hydride bed, the use of heat exchangers is the most common way to solve heat and mass transfer problems in metal hydride beds due to the simplicity and efficiency. The design of cylindrical-shaped reactors makes it possible to optimize both the number of tubes and to select a specific structure of heat transfer elements. In this regard, there are extensive studies relating to the selection and optimization of the heat exchangers configuration [20,21]. The indisputable advantages of heat exchangers are the flexibility and great variability of the geometries of heat exchangers and heat transfer surfaces, as well as the possibility of supplying both heated and cooled liquid through the heat exchanger tubes.

A large number of solutions for the problem of poor heat and mass transfer in metal hydride layers have been proposed. The most common method of solving this problem is the use of heat exchange systems inside the reactor. Many scientific papers present mathematical calculations, as well as the results of simulation and design optimization of metal hydride reactors, performed in order to optimize the number of heat exchange elements and determine the best configuration of a metal hydride bed. A large number of review works have also been carried out, in which some thermal management solutions and configurations of metal hydride reactor with heat exchangers were considered [21,22,23,24,25,26,27,28].

According to the studies carried out, it is possible to sum up the number of reports and articles aimed at solving the problems of heat transfer in metal hydride reactors (Figure 1).

Currently, 3D technologies are receiving a lot of attention as well. Additive manufacturing technologies can be used to create high-efficiency heat exchangers designed for installation in metal hydride reactors. Additive manufacturing makes it possible to produce small parts with exceptionally complex shapes, as well as internal structures and matrices that are impossible with any other method of manufacturing. Additively manufactured aluminum heat exchangers already exist and are used in many applications [29,30]. Pure copper 3D printing technologies are also being actively developed nowadays [31,32]. Thus, using 3D printing technologies, the weight of the product is reduced, the efficiency is increased, and the manufacturing and assembly time is reduced, since multi-component assemblies can be 3D printed as a single unit. This method facilitates a more efficient utilization of relatively expensive metals because only the amount of material needed for each part is used. This makes it possible to manufacture heat exchangers of various shapes with only a slight increase in the mass of the metal hydride reactor [33,34].

This work discusses recent advances in improving and optimizing the geometry of heat exchangers, heat transfer surfaces and metal hydride bed. Given the importance of developing the optimal geometry of the heat exchanger and determining the metal hydride bed configuration, a review of existing solutions can be useful for generalizing the developments of other scientific groups and implementing their own technical solutions.

## 2. Heat Transfer Issues

Three major heat transfer problems can be distinguished in solid-state hydrogen storage and purification systems. The first is the thermal effect at the gas–solid interface that occurs during hydrogen sorption. The second problem is poor heat removal (supply) in the process of hydrogen sorption (desorption) due to the low thermal conductivity of the hydrogen storage material. And the last one is the influence of the temperature of the hydrogen storage material particles on the equilibrium pressure.

Low sorption and desorption rates are among the major drawbacks of existing reactors and compressors based on metal hydride storage materials. The irregular shape of the particles leads to additional thermal (contact) resistance between them. Fine powders (average particle size of the order of 10 microns) used as hydrogen storage materials have low thermal conductivity [35]. At present, solving the problem of increasing the thermal conductivity in low-temperature metal hydride reactors is one of the important studies [36].

In addition, due to the large local thermal effect of the hydrogen sorption/desorption reaction (about 20–80 kJ/mol H_2_) by the hydrogen storage material, the temperature increase can be so large that the process stops. This phenomenon is called the heat and mass transfer crisis and is characterized by a sharp decrease in the hydrogen consumption at the inlet to the metal hydride reactor, which leads to a decrease in its operating speed [21,37,38,39]. In the case of high-temperature metal hydrides, the powder particles in the metal hydride bed can also sinter at high internal temperatures and lose their ability to store hydrogen. However, many hydrogen storage materials require heating to a certain temperature, determined by chemical thermodynamics, to activate the sorption/desorption process. Without a sufficient heat supply, the evolution of hydrogen will stop due to the lower temperature in the metal hydride layer. In this regard, the development of an efficient system for supplying and removing heat in metal hydride reactors is an urgent task.

In general, excluding the use of phase change materials, there are two ways to improve heat transfer in metal hydride hydrogen storage systems: changing the thermophysical properties of the metal hydride layer in order to increase their effective thermal conductivity (for example, mixing and/or sintering metal hydride powder with thermally expanded graphite or powders of highly thermally conductive materials, such as aluminum or copper) or the development of the heat exchange surface area by the introduction of additional heat exchangers and heat-conducting surfaces into the metal hydride reactor (fins made of aluminum or copper, the addition of several tubes and channels for the flow of the coolant inside the metal hydride layer, thinning of the hydrogen-absorbing material layers, etc.).

## 3. Metal Hydride Reactor Types and Shapes

The metal hydride reactor shape influences the convenience and possibilities of variation and control of heat in a metal hydride bed. The common form of a metal hydride reactor, which provides high hydrogen pressure and allows the metal hydride to be placed together with a heat exchanger, is a cylindrical-shaped reactor. Cylindrical-shaped reactors provide a high operating pressure and facilitate radial heat and mass transfer in the metal hydride layers, ensuring uniform absorption/release of hydrogen by the metal hydride bed [40,41]. Moreover, reactors of this shape are relatively easy to manufacture. Cylindrical-shaped reactors can be divided into two types: a chamber (tank type) reactor (Figure 2a), in which hydrogen is supplied directly to the metal hydride bed and diffuses through it to the bottom, as well as a tubular reactor, in which the metal hydride is located in the area between the cylinder wall and the porous metal filter tube, and as a result, hydrogen is supplied to various parts of the metal hydride bed (Figure 2b).

Tubular reactors are often connected to each other in blocks, allowing you to store a large amount of hydrogen. While tubular reactor configurations are widely used for both metal hydride storage systems and metal hydride compressors, chamber reactors are used in hydrogen storage equipment and systems. It is also worth noting that some papers argue for a more efficient reactor design with an elliptical cross section instead of a round one in terms of compactness and low hydraulic losses [42].

To ensure sufficient heat transfer in the radial direction, the outer diameter of a single tubular reactors is usually about 30 mm or less. Therefore, the length of the metal hydride reservoir is often increased in order to achieve an acceptable amount of metal hydride. In practical applications, long reactors with a small cross-sectional area are used. Tubular reactors can be used as a single unit for laboratory testing and smaller applications [42,43,44], as well as in modular designs in applications requiring a larger capacity [45,46]. Recent studies of such systems have focused on various pipe configurations and geometries.

For a chamber tank, one of the advantages is the ability to store a larger amount of hydrogen storage materials compared to a tubular tank, and as a result, such reactors have a large capacity. However, the performance of a given metal hydride reactor is highly dependent on the number and arrangement of heat and mass transfer elements, which must be carefully designed to meet the requirement for good performance. It should be noted that the same porous filter tube for hydrogen supply can be installed in chamber reactors with heat exchangers, as in tubular reactors. The filter was installed at a distance from the walls of the tank, and insufficient heat transfer in the radial direction with large tank sizes is compensated for by the presence of a heat and mass transfer system. Such vessels can be used for both storage and compression of hydrogen [47,48].

In addition to the cylindrical-shaped design, spherical [49] and disk [50,51] types of reactors are also considered (Figure 3).

For spherical-shaped metal hydride reactors, they have a high capacity for metal hydride powder, but it is quite difficult to design heat exchangers for this type of cylinders. However, according to the simulation results presented in [49], a spherical-shaped reactor has better efficiency in combination with a phase change material than cylindrical-shaped reactors. For disk-shaped metal hydride reactors, the metal hydride bed is flat, and therefore, a larger heat transfer area is reported than for other shapes. For such a thin layer of metal hydride, fast reaction kinetics are observed; however, such reservoirs have a small capacity, and due to their design features, it is extremely difficult to increase productivity by increasing the number of reservoirs.

Thus, tubular type reactors are preferred for use in hydrogen compression systems where capacity is a key parameter, while chamber-type reactors are best used for hydrogen storage systems. As for cylinders of disk and spherical type, their use is limited by design features and kinetic measurement.

## 4. Metal Hydride Bed Modification

The effective thermal conductivity is increased by increasing the contact area between the particles. Thus, the metal hydride particles must be densely packed (compressed) for better thermal conductivity. However, despite the positive effect of heat transfer, too much compression leads to a significant reduction in the ability to store hydrogen [52]. Nevertheless, compacts are quite often used in research, since loosely packed particle beds have a much lower heat transfer efficiency.

To increase the effective thermal conductivity of the metal hydride powders, many studies use the compacts from metal hydride powders, obtained by pressing hydrogen storage material with additives. This method was firstly proposed by Ron M. et al. with aluminum powder [53,54,55], Kim K. J. et al. with copper and tin powder [56,57] and Klein H. P. et al. with expanded graphite [58,59,60].

Several scientific works have shown that copper coating of hydrogen storage material particles has a high efficiency. Thus, Kim K. J. et al. [56] considered coupled metal hydride reactors with Ca_0.4_Mm_0.6_Ni_5_ (Mm = Mischmetal). To improve heat transfer in the metal hydride bed, the authors used copper-coated particles, which were then pressed into porous metal hydride compacts. The addition of Al and Cu requires a complex chemical treatment of the powders before their use in the metal hydride reactor. Furthermore, the addition of these materials significantly increases the parasitic thermal mass of the metal hydride bed as well and, therefore, introduces significant thermodynamic penalty on performance. However, it has been shown that the compact copper-coated powder has a thermal conductivity 10–50 times higher than that of the free hydride powders [57]. Deng C. et al. [61] proposed a simpler and more efficient method of copper plating of the metal hydride powder surface to improve the thermal conductivity of the reactor bed. A solution of copper II sulfate (CuSO_4_) was used as a modifying reagent, and hydrogen fluoride was used as a catalyst. The authors point out that the modified alloys have a lower contact resistance, which leads to improved electrochemical characteristics. Romanov I. A. et al. [62] studied a composite based on an activated intermetallic compound La_0.9_Ce_0.1_Ni_5_ and copper powder for hydrogen storage. The authors confirmed that the decrease in the temperature relaxation time and the equilibrium pressure of hydrogen absorption and desorption depends on the improvement of the thermal conductivity of the metal hydride to which Cu has been added. In addition, the enthalpy and entropy of hydrogen absorption by the composite was significantly lower compared to the pure intermetallic compound. Atalmis G. et al. [63] performed copper plating of LaNi_5_ to increase the thermal conductivity. It was observed that the copper coating of the metal hydride increased the thermal conductivity by 500–750%.

One of the most well-known additives for improving the heat transfer properties of metal hydride bed is the addition of expanded natural graphite (ENG) and formation of metal hydride compacts (Figure 4a).

Heated expanded graphite is mixed with metal hydride particles to produce compacts. Such compacts have improved thermal conductivity without large parasitic heat losses (Figure 4b) [64].

Pentimalli M. et al. [65] introduced the active metal phase into the silica matrix, after which the resulting composition was ball milled with graphite. Experimental data obtained by measuring the thermal conductivity showed that the use of powdered graphite leads to a fairly linear increase in the thermal conductivity of the metal hydride–silica composite.

A comparison of expanded graphite and metal foam compacts when added to metal hydrides showed that the effective thermal conductivity was several times higher with expanded graphite compacts, and the decrease in hydrogen sorption rate was insignificant [60]. However, one drawback has been found in that swelling of the hydride can lead to the decomposition of the compacts or deteriorate the contact between compacts and wall. Chaise A. et al. [66] observed a remarkable improvement in the radial thermal conductivity of compacts obtained from ball-milled magnesium hydride powder mixed with expanded graphite (10% wt.). The results showed that the thermal conductivity came to around 7.5 W/(mK). However, in the axial direction the thermal conductivity was not so good compared to the radial one. In addition, because of a reduced difference in densities between ball milled MgH_2_ powder and expanded graphite, a nice homogeneous mixture is easily obtained.

The authors [67] report that the combined use of expanded graphite and molten Mg provides a thermal conductivity of about 47 W/(mK) in the radial direction. The authors also showed that the value of effective thermal conductivity must lie between 10 and 20 W/(mK) in order to achieve 1 wt% storage of hydrogen each minute. Madaria Y. and Kumar E. A. [68] showed that the maximum values of effective thermal conductivity increased to 4.7 W/(mK) for a compact with graphite flakes and up to 6.8 W/(mK) for a compact with graphite and copper wire, which is four to six times higher than for a loosely packed La_0.8_Ce_0.2_Ni_5_ powder (1.3 W/(mK). Delhomme B. et al. [69] proposed a method for improving heat transfer in large Mg/MgH_2_ storage metal hydride reactors by creating metal hydride/ENG compacts. In a recent work, Singh U. R. and Bhogilla S. [70] determined using numerical simulation that during absorption, the maximum average temperature of the metal hydride bed without expanded graphite addition can reach 345.13 K, while in the metal hydride compact with thermally expanded graphite, a temperature of 337.37 K is predicted. The authors also conclude that the addition of a heat exchanger and raising the coolant temperature to about 298 K will significantly increase the absorption time in both cases. Dieterich M. et al. [71] investigated the hydrogenation performance of Hydralloy^®^ C5-based metal hydride composite compacts with the addition of 10 wt% ENG. The authors found that after about 250 cycles, a fixed effective thermal conductivity of 10–15 W/(mK) is achieved. Metal hydride compacts with the expanded graphite are also considered in the works of Park C. S. et al. [72] and Bao Z. et al. [73,74]. It should also be noted that in addition to expanded graphite, other carbon additives, such as nanotubes, can be used as a material that can improve thermal conductivity of metal hydride bed [75].

Another studied method for increasing heat transfer in a metal hydride bed is the insertion of solid disordered matrices. Most often, foam metals are used in this role (Figure 5).

Foam metal is characterized by a large surface area with a reduced volume, low density (porosity above 90–95%) and good thermal conductivity over 100 W/(mK). The high strength structure is also one of the main characteristics that make these solutions good options for increasing the effective thermal conductivity of the metal hydride bed. The most common material is aluminum foam, which combines lightness and high thermal conductivity [76,77,78,79,80,81,82].

Suda S. et al. [83] investigated the performance of a metal hydride bed packed with aluminum foam. The authors found that the heat transfer of the hydride bed with metal foam was 5–7 W/(mK), which is 10 times higher than that of general hydride beds. It has also shown that an aluminum foam matrix can prevent powder from flowing and stacking in the bed. Laurencelle F. et al. [81] validated their 1D model for a small and large metal hydride reactor equipped with aluminum foam. The authors concluded that a reactor without a heat exchanger should be of small diameter (less than 8 mm) to ensure a fast sorption/desorption reaction. However, if the metal hydride bed is packed with aluminum foam, this limit can be extended to a diameter of 60 mm. The authors also found that a foam grade of <20 pores per inch is not effective due to poor heat transfer. These results were further confirmed by Mellouli S. et al. [84] where a 2D mathematical model of a hydride reactor was considered with various foam metals, including zinc foam, copper foam and aluminum foam (Figure 6).

According to the results obtained, the most effective material considered is aluminum foam. The influence of the volume fraction of the foam metal on the hydrogenation time was considered in the works of Tsai M. L. and Yang T. S. [85], Wang H. et al. [86] and Ferekh S. et al. [87]. Thus, Wang et al. showed that the addition of 10 vol.% aluminum foam maximizes the accumulation of hydrogen by the LaNi_5_-based hydride with a filling time of about 3 min.

There are several publications that report the use of a copper wire matrix. Nagel M. et al. [88] investigated a MmNi_4.46_Al_0.54_ metal hydride bed packed with a copper wire net matrix. It was found that the heat transfer coefficient of the hydride bed with the addition of a three-dimensional copper network complex structure was about 7 to 12 W/(mK), while for the metal hydride bed with aluminum foam, this parameter was 5–7 W/(mK). A method involving the use of ordered matrices has great potential as well. The ordered matrices can be optimized for a particular metal hydride reactor and allow more precise temperature control. However, the production of such matrices by conventional production methods is extremely difficult, and therefore, less complex and more profitable solutions are used.

Table 1 provides a brief summary of the considered methods of metal hydride bed modification.

Despite a significant improvement in effective conductivity, the use of foam metals contributes to a decrease in the hydrogen storage capacity, hinders the diffusion of hydrogen in the metal hydride layers, and weakens the mass transfer process. In this regard, alternative solutions are being considered, including the use of cooling tubes, fins and external jackets. The use of these heat exchangers is the most promising choice; however, their properties and the effect on thermal conductivity in the metal hydride bed are determined by their design.

## 5. Heat Exchangers

In recent years, methods for optimizing the geometry of heat exchangers have been actively developed as an effective way to solve heat transfer problems in metal hydride beds. However, since experimental studies of the processes occurring in hydrogen storage systems require quite complex facilities and involve high material costs, numerical simulation studies are relevant [94]. In many articles, various software systems and environments, as well as mathematical software packages, serve as tools for heat transfer simulation in a metal hydride reactor. These tools include computational fluid dynamics (CFD)-based analysis software Ansys Fluent^®^ 19.9 and other versions [95], finite element method (FEM) software COMSOL Multiphysics^®^ 2005, 2011 and other versions [96,97,98,99,100,101,102], open source CFD-toolbox OpenFOAM^®^ [103], advanced process modelling platform gPROMS^®^ [20,104,105] and another software.

The use of heat exchangers is a common way to solve heat transfer problems in metal hydride reactors. The design of cylindrical reactors makes it possible to optimize both the number of tubes and to select a certain structure of heat exchange surfaces, such as fins. In this regard, there are extensive studies relating to the selection of the configuration of the heat exchanger.

### 5.1. Design and Number of Cooling Tubes

Cooling tubes are designed to circulate the coolant during hydrogen sorption. When designing metal hydride reactors, research papers consider straight tubes, U-tubes and helical coils. Much research is focused on optimizing the number of tubes needed for effective temperature control inside the reactor. Straight cooling tubes [106], helical coils [107,108,109] and U-shaped tubes [110,111] are considered in the scientific literature. Overall, it has been observed that the use of a helical coil heat exchanger increases the heat transfer coefficient more significantly than a similarly sized straight tube heat exchanger. A helical coil heat exchanger for metal hydride reactor was investigated by Mellouli S. et al. [112]. The authors considered the influence of such important parameters as flow mass, temperature of the cooling fluid, applied pressure and hydrogen tank volume, and they reported that the cylindrical-shaped metal hydride reactors charge/discharge time is significantly reduced when using a helical coil heat exchanger.

Later, Mellouli S. et al. [113] proposed a simulation investigation on three metal hydride reactors with different heat exchangers. The authors performed simulations and compared the reactor with a helical coil heat exchanger considered in their previous work (configuration 2) with two other configurations, assuming a water jacket design outside the wall (configuration 1) and combining both heat exchangers together (configuration 3). The simulation results showed that the combination of both heat exchangers resulted in a 30% improvement in reaction time compared to the cooling jacket alone (Figure 7).

In this case, the configuration with an internal helical coil heat exchanger shows better results compared to the external heat exchanger, since the heat transfer path of the internal cooling scheme was smaller than that of the external one.

Wang H. et al. [114] proposed the use of aluminum foam in a helical tube metal hydride reactor based on numerical simulations. The results showed that the addition of 5% aluminum foam is optimal, and the convection coefficient in the cooling tube, equal to 1000 W/m^2^K, is sufficient to bring the metal hydride to saturation in 3 min. Sekhar B. S. et al. [115] found that the dynamics of hydrogen absorption improved in the order: internal straight tube heat exchanger < external cooling/no fins < internal helical coil heat exchanger ≈ external cooling/transverse fins. This indicates the advantage of external cooling for the metal hydride reactor of the studied geometry and size chosen by the authors. In the mathematical model for three configurations with straight tube (Case A), finned tube (Case B) and helical tube (Case C), Wu Z. et al. [116] compared heat and mass transfer in a magnesium metal hydride reactor. As a result of the modeling, it was found that the best heat and mass transfer process showed the helical coil heat exchanger (Case C). The highest desorption rate was also observed for the helical coil heat exchanger. Case B had worse heat and mass transfer characteristics and a worse desorption rate than Case C, but better than Case A. The authors found that the pitch of the helix has a direct effect on improving heat and mass transfer characteristics and the desorption rate. Later, Wu Z. et al. [117] showed a significant improvement in desorption performance by incorporating a helical coil heat exchanger to a metal hydride reactor with a Mg_2_Ni bed (Figure 8).

Helical tubes show great efficiency when introduced into metal hydride reactors. They were used to determine the effect of various operating parameters, such as the convection coefficient [117], coolant temperature and hydrogen pressure [118,119]. Experimental studies and theoretical calculations using simulations show that the listed operating parameters have a greater influence than the geometric parameters of the helical coil heat exchanger. However, even though the helical coil heat exchanger shows better efficiency in improving the heat and mass transfer properties of the metal hydride cylinder in comparison with a straight tube, the metal hydride bed temperature in the central area of the helical coil heat exchanger remains too high to hinder the hydrogen absorption reaction because of the low thermal conductivity coefficient of metal hydride, which means the heat cannot reach the cooling boundary quickly through heat conduction.

The cylindrical-shaped reactor with a helical coil heat exchanger structure could also be developed by combining the optimization ideas of other models. Thus, Visaria M. et al. [118,119] studied the thermal performance of a helical tube heat exchanger for hydrogen storage systems using a high-pressure metal hydride. The authors used a model of a metal hydride reactor based on Ti_1.1_CrMn operating at a hydrogen pressure of 280 bar and a hydrogenation reaction temperature 50 °C. The main goal of the study was to achieve the design of the heat exchanger with the maximal efficiency of hydrogen storage (by reducing the volume of the heat exchanger). Therefore, the researchers rejected the use of fins and used the helical coolant tube with a central return tube instead. AISI 316 stainless steel was used as the tube material. Comparison of tests with and without coolant supply showed a 75% reduction in time required to fill the metal hydride reactor with hydrogen when adding a heat exchanger, while the heat exchanger occupies only 7% of the volume of the reactor. Simulation results also confirm the high efficiency of the heat exchanger. A 25% reduction in hydrogen sorption time by the storage material was achieved by locating the central return tube inside the heat exchanger [120].

An interesting modification of the metal hydride reactor was proposed by Feng P. et al. [121]. In their work, using the developed technique, the helical coil tube was optimized, which led to an increase in the gravimetric exergy output rate from 198.4 to 255.4 W/kg. The optimal structure was a U-shaped double-spiral tube with a diameter of 30 mm. In addition, areas where heat transfer was least efficient were eliminated by the bed geometry modification. The final design of the powder bed with helical tube based on optimal design methodology constitutes a significant increase in metal hydride reactor efficiency.

Tong L. et al. [122] considered a metal hydride reactor with a LaNi_5_ bed equipped with straight tube or helical coil heat exchangers, as well as their combinations. The authors found that a helical coil heat exchanger shows better efficiency than a straight one. The simulation results also showed that the combination of two serpentine heat exchangers is the most efficient method presented in this study. A metal hydride reactor without a heat exchanger, with a straight tube heat exchanger and with a helical coil heat exchanger requires 1531, 1012 and 419 s, respectively, to reach 90% of the maximum value of the hydrogen capacity. Comparing the helical coil heat exchanger and combinations of different configurations, the authors found that the difference in average temperature between the three cases is not obvious in the first 100 s; then, however, the heat exchanger in the form of a combination of two helical coil heat exchangers shows better heat transfer characteristics than others. Overall, the H_2_ absorption rate increases when introducing a heat exchanger in the order: straight tube < helical tube < helical tube and straight tube < dual helical tube.

Integrating the design concept of a straight finned tube and helical tube, Dhaou H. et al. [123] reported a finned helical coil heat exchanger design. This heat exchanger consists of a helical tube made of stainless steel and copper fin coils. Helical tubes with copper fins were inserted into the innermost helical coil heat exchanger. Continuing the development of this concept, Mellouli S. et al. [124] investigated a helical design ribbed on both sides and four different configurations. It was observed that the integration of dual helical tubes equipped with fins exhibited a significantly faster rate compared to the other solutions. Thus, about 420 s is required to fulfill the 90% storage need. The metal hydride reactor equipped with dual helical heat exchangers without fins was about 80 s slower. The metal hydride reactor with only one helical tube took the longest time of 1400 s to reach this value. The increased contact area between the heat exchanger and the metal hydride packings is responsible for the improved sorption rate.

These kinds of cooling tubes could also be used to design mini-channel reactors. Ma J. et al. [125] presented a finned multi-tubular heat exchanger to enhance the heat transfer in the metal hydride bed. Li H. et al. [126] proposed a new helical mini-channel reactor to increase the heat transfer efficiency. The structural parameters of helical tubes were also investigated. The authors have shown that the radius (minor and major) and the axial step of the heat exchanger affect the reactor performance. The minor radius is defined as the radius of the tube, and the major radius is the distance from the center of the metal hydride layer to the helical tube. A larger heat transfer area can be obtained by increasing either radius. Increasing the axial step leads to a decrease in the contact area and, consequently, a deterioration in the sorption/desorption reaction kinetics. Therefore, the smallest axial step and the largest and major radius are the best configurations for improving heat and mass transfer. This has been confirmed by other similar studies [114,116,127]. 

The elliptical tube duplex reactor (DHER) was proposed by Wang D. et al. [128]. The authors modeled heat transfer in LaNi_5_ powder using this type of reactor and also considered five other typical reactor models, including the conventional simplex straight tube reactor (SSR), simplex helical elliptical tube reactor (SHER), DHER, triplex helical elliptical tube reactor (THER, design inspired by deoxyribonucleic acid structure) and duplex helical circular tube reactor (DHCR). The number of such helical tubes in the reactor was also increased from one to four. The results of comparing these designs showed the outstanding performance of DHER, which can reduce the absorption time by 14% and desorption time by 15% compared to SHER, and the maximum axial temperature difference can be reduced by 5.1 K and 5.6 K in absorption and desorption, respectively. It was found that the optimal installation angle of heat exchange tubes (position parameter, *β*) was determined to be *β* = 180°. The authors also note that results obeyed the sequence *Dc* > *A* > *B* > *Pt* > α, where *Dc* is spiral diameter, *Pt* is pitch, *A* is elliptical major axis, *B* is elliptical minor axis and α is the tilt angle. 

Another modification of tube geometry was proposed by Larpruenrudee P. et al. [129]. Authors numerically investigate metal hydride reactor with Mg-based storage material. A semi-cylindrical helical coil was proposed as heat exchanger. As a result, the hydrogen sorption time is reduced by 60% compared to a simple helical tube design. The authors found that reducing the axial pitch of the spiral also reduced the rate of hydrogen sorption.

Wang D. et al. [130] conducted a comparison of various tube designs and proposed the concept of a radiation tube (Figure 9).

The authors found that the convergent-divergent tube had a slight performance improvement over the straight tube. At the same time, helical tubes lower the temperature of the metal hydride bed and increase the reaction rate more significantly, but the difference between these helical tubes is not very noticeable. The radiant tube had the best performance, making it possible to reduce the reaction time of hydrogen absorption by 52% and 37% compared to the straight tube and helical tube, respectively. Moreover, the reactor with an embedded radiant tube behaved most favorably in terms of bed temperature distribution. Based on these results, the authors optimized the parameters of the radiant tube and proposed two heat exchangers that evidently outperformed all the other metal hydride reactors, reviewed by the authors.

The embedding of annular cooling tubes is one more alternative to the straight tube and helical coil geometry. A metal hydride reactor with annular ring-shaped tube heat exchanger was introduced by Gopal M. R. et al. [131]. After conducting numerical simulations, the authors concluded that in order to obtain good rates of heat and mass transfer, the thickness of the hydride layer should be as small as possible, and the thermal conductivity of the layer should be as large as possible. A more uniform hydrogenation process is achieved by connecting the inner coaxial tube to the heat exchanger. Kikkinides E. et al. [132] showed that the design of an annular heat exchanger and a cylindrical inner tube reduces the sorption time by 60% at full charge. In another study [20], the authors showed that simultaneous external and internal cooling has a negligible improvement on the charging rate of the metal hydride reactor. This is due to the fact that the additional cooling had no effect on the cooling of the system as the ring cooling was able to provide a uniform radial heat transfer rate. 

As part of the existing research related to optimizing the geometry of heat exchangers, the U-shaped cooling tubes could also be used. A performance comparison of a LaNi_5_ metal hydride reactor with a single finned tube heat exchanger and a finned double U-shaped tube heat exchanger was provided by Bai X. S. et al. [111] in their simulation work. The results showed that the charging time for the U-tube model was 1200 s, while the single finned tube model required 2800 s. However, the authors have developed a more advanced heat exchanger design with loop-type fins and an external heat exchanger. The reactor model used four tubes for supplying hydrogen. Circular fins were embedded into the reactor with central cooling tube and cooling jacket. Thus, fins were divided into the inner fins installed on the straight tube and the outer fins installed on the water jacket side. The charging time of the proposed hydride reactor for reaching 90% hydrogen saturation has decreased by approximately 57% and 82% relative to hydride reactors with a single finned tube heat exchanger and a finned double U-shape tube heat exchanger, respectively. In addition, the authors also reviewed three configurations of metal hydride reactors with 7 fins with the fin thickness of 1 mm, 14 fins with the fin thickness of 0.5 mm and 35 fins with the fin thickness of 0.2 mm. It was observed that increasing the number of fins with a small thickness can increase the hydrogen absorption reaction rate (Figure 10).

Singh A. et al. [133] and Mahmoodi F. et al. [134] suggested using a U-shaped finned heat exchanger in a solid-state hydrogen storage system as well. In their studies, a good heat removal effect was obtained. In this study, the authors found that increasing the tube diameter improves the heat transfer performance of the second stage. The authors also considered the effect of the fins number and their thickness on the properties of the metal hydride reactor.

In general, the key task for metal hydride bed design is to improve the thermal conduction area and reduce the thermal resistance. Thus, modifications to the tube geometry and coolant path discussed above may be a solution for this purpose. Increasing the number of cooling tubes seems to be another way to meet this goal. Raju N. N. et al. [135] proposed a relationship for the optimal number of heat exchanger tubes in a metal hydride reactor. Using 3D mathematical models, the authors studied various metal hydride reactor configurations for a 50 kg La-based metal hydride bed. A reactor with a nominal diameter of 6 inches and 99 heat exchanger tubes was found to exhibit optimum performance (Figure 11).

Raju N. N. et al. in the other work [136] tested their metal hydride reactor with 99 embedded cooling tube and filled with 40 kg of LaNi_4.7_Al_0.3_. The authors also investigated the influence of various operating parameters, such as fill pressure, coolant flow rate and absorption temperature, on the hydrogenation reaction. It has been found that the optimum supply pressure for hydrogen absorption is in the range of 10–15 bar. The optimum temperature for desorption of hydrogen should be around 80–90 °C. The optimum flow velocity for coolant was observed in the range of 20–30 L per minute.

Mohan G. et al. [95] investigated the effect of various material properties on the performance of a metal hydride reactor equipped with multiple cooling tubes. It was found that the process of hydrogen absorption is most affected by the absorption rate constant, activation energy and thermal conductivity. By studying the influence of the cooling tubes number, it was observed that the best performance was achieved when the number was 85.

Anbarasu S. et al. [137] carried out studies with different numbers of cooling tubes in a metal hydride reactor. The authors considered 2D and 3D models of symmetrical reactors with different numbers of tubes, equal to 24, 36, 48, 60 and 70 pieces. The inner diameter of the reactor was 103.4 mm with a reactor length of 160 mm. The outer diameter of the cooling tube was 6.35 mm with a wall thickness of 1 mm. The heat transfer fluid (HTF) tubes were located at diameters of 36, 58 and 80 mm from the central tube with different packing densities (Figure 12). It was found that the maximum effect in improving the heat and mass transfer characteristics was achieved by increasing the number of tubes up to 60. An extremely slight difference was observed between the configurations with 60 and 70 embedded tubes.

Keith M. D. et al. [110] simulated several configurations of a Ti_1.1_CrMn-based metal hydride bed with a single coolant tube passing through the reactor several times. The tubes and fins were supposed to be aluminum with a wall thickness of 1 mm. The tubes were evenly distributed around the central part. The configurations that were tested included 2-, 3-, 4-, 5-, 6-, 9-, 12-, and 18-pass heat exchangers (Figure 13).

The authors also consider configurations with twelve fins attached to the cooling tubes. The fin aspect ratio (length divided by width) was the criterion used to determine the optimum number of fins for charging. It was found that increasing the number of finned tubes up to five reduced the reactor charging rate, with the effect of the speed reduction becoming insignificant as this parameter was increased. Adding fins compared to tube-only configurations achieved a significant reduction in charging time of 56% to 68%. However, the authors concluded that above nine passes of the cooling tube in the metal hydride bed were unreasonably high for a hydrogen fueling application. The simulations carried out by Boukhari A. et al. [138] took less than 3 h to touch saturation value, while the other configuration with four or five cooling tubes took about 5 h to achieve the same capacity. These results indicated that the increasing the number of cooling pipes can significantly speed up the heat transfer and reaction rate in the metal hydride bed.

Muthukumar P. et al. [139] considered a reactor for a MmNi_4.6_Al_0.4_ metal hydride bed with multiple cooling tubes. The cooling tubes are placed at two radii of 10 mm and 20 mm, respectively. The authors suggested that the addition of tubes beyond 20 does not have significant effect on the absorption time. The optimization results showed that the number of cooling tubes was the most crucial factor influencing the thermal behavior and absorption rate. Based on the results obtained by the authors, the optimal number of cooling tubes was determined. It was found that best and so called “optimum” performance shows a reactor with a diameter of 50 mm with 20 cooling tubes, and for large-scale hydrogen storage in a reactor with a diameter of 300 mm, the best sorption efficiency will be observed with 48 cooling tubes. Later, Gkanas E. I. et al. [140] also presented several different metal hydride reactor geometries based on three different materials (LaNi_5_, MmNi_4.6_Al_0.4_ and AB2-intermetallic) (Figure 14). The authors found a dependence of the saturation time on the number and location of tubes similarly to that presented in work of Muthukumar P. et al. [139]. Bao Z. et al. [141] proposed metal hydride reactor equipped with five cooling tubes. The influence of turbulence through a numerical simulation was investigated.

In the scientific literature, capillary reactors with a large number of tubes were also considered. This type of the metal hydride bed was discussed by Dehouche Z. et al. [142], as well as Willers E. and Groll M. [143]. The same concept of a metal hydride reactor was proposed by Linder M. et al. [144,145]. The authors proposed a reactor equipped with 372 stainless steel tubes with an inner diameter of 1.4 mm. The coolant flows through the tubes that are embedded into metal hydride bed.

The study of the influence of the location of the cooling tubes in the metal hydride bed was considered in several works. Liu Y. et al. [146] performed a numerical simulation to evaluate the influence of the location of the cooling tubes relative to the center of the metal hydride reactor on the hydrogen accumulation rate. For convenience, the values of the saturation time were normalized, and the value equal to one was taken as a configuration with one straight cooling tube located in the center of the metal hydride reactor. The authors note that an increase in the number of cooling tubes near the central part of the metal hydride reactor leads to a slight decrease in saturation time due to increasing the heat transfer area from the tubes to the metal hydride bed. Increasing the distance of the cooling tubes relative to the center of the metal hydride reactor leads to a sharp reduction in the saturation rate. Then, the saturation rate increment slows down and reaches an extremum point, where the sum of the distances from each part of the reactor to the cooling boundary has reached a minimum. After that, a further increase in the distance between the central part of the metal hydride reactor and the cooling tubes prevents the heat transfer process and reduces the reaction rate. For the metal hydride reactor chosen by the authors, the optimal pipe distance value in the configurations were 50 mm, 53 mm and 60 mm, respectively.

Dubey S. K. and Kumar K. R. [147] also performed a study of the number of heat transfer tubes and aspect ratio on energy desorption and heat transfer from a metal hydride bed. The authors found that the number of tubes for a reactor with an aspect ratio of 0.5 (diameter of metal hydride bed is 80.7 mm), 1 (diameter of metal hydride bed is 99.1 mm) and 2 (diameter of metal hydride bed is 122 mm) was 32, 48 and 72, respectively.

### 5.2. Design and Number of Fins

The use of internal or external fins is a common method for improving heat transfer in metal hydride bed as well. External fins are used to provide natural convection. Obviously, the external fins did not show an obvious heat transfer advantage over cooling pipes, and the performance of this solution is inferior to internal fins. Internal fins provide a larger heat transfer area. Typically, internal fins or disks are used in combination with heat pipes. This solution not only greatly increases the heat transfer area and separates the metal hydride bed into separate domains, but it also provides better local thermal conductivity. The fins can be of different shapes and positioned in different ways in a cylindrical-shaped metal hydride reactor, allowing researchers to design a huge variety of different configurations. Askri F. et al. [148] proposed an external finned design and internal heat exchanger based on cooling pipe with fins. The authors found that the time required for 90% hydrogen absorption was reduced by almost 80% for a reactor equipped with a finned tube heat exchanger. In addition, external fins provide better heat transfer in a metal hydride bed compared to a metal hydride reactor design without fins. However, compared to the internal heat exchanger, the effect of the external fins on the metal hydride reactor performance was less significant. Kaplan Y. [149] considered a metal hydride reactor with external circular fins as well. In his work, this design was also compared with a simple metal hydride reactor without a heat exchanger and a reactor equipped with an external jacket in which the cooling liquid was circulated. Among the considered reactors, the best results in terms of charging time and temperature linearity are obtained by a reactor equipped with an external jacket. Nevertheless, external fins also demonstrate improved heat and mass transfer [150].

External fins can also be attached to tubes so that the circulating cooling liquid releases heat via convective heat transfer. Thus, Chung C. A. et al. [151] conducted an experimental study of the effect of heat pipe embedding on the performance of a metal hydride reactor using LaNi_5_ as the storage media. To improve heat transfer in the metal hydride bed, both internal and external transverse fins attached to the cooling tube were used (Figure 15).

In addition, the reactor was placed into the water bath. The authors found that for the considered metal hydride reactor, the addition of a finned cooling tube reduced the absorption time by more than 50%, with a 10 atm hydrogen supply pressure, and the desorption time by 44%.

The addition of internal fins are the more common solution due to greater efficiency compared to external ones. Manai M. S. et al. [152] conducted a study of the sorption and desorption reaction dynamics for four different configurations of metal hydride beds (Figure 16).

A Ti-Mn alloy was used as a hydrogen storage material. The authors have developed a numerical model for heat transfer and hydrogen sorption/desorption rates simulation. According to the results of the analysis, the 3rd and 4th configurations provided better heat transfer efficiency, absorption and desorption rates, and increased hydrogen storage density. The authors point out that the 3rd configuration would give an additional advantage because of its simplicity and reduced manufacture cost.

A combination of external and internal fins was used in a metal hydride reactor model proposed by Nyamsi S. N. et al. [153]. The authors performed a numerical investigation of the dehydrogenation performance of a cylindrical-shaped metal hydride reactor filled with 300 g of milled Mg_90_Ti_10_ + 5 wt% C. Using basin-like fin heat exchanger with absence of internal cooling tubes, the authors achieved a power output rating from 100 to 250 W with a nearly constant flow rate.

Many works present branched structures and fins with a large number of holes. The advantage of such complex configurations is a larger contact area, as well as a lower weight and parasitic volume being occupied. The combination of plate fins with cooling tubes has been proposed by Gkanas E. I. et al. [154] (Figure 17).

The authors conducted a parametric study of a system based on MmNi_4.6_Al_0.4_, in which the thickness and number of the fins, as well as the heat transfer coefficient, were varied. The authors found that the optimal number of fins for a reactor with 13 kg of metal hydride powder was 65, and the optimal thickness of fins was from 5 to 8 mm.

Afzal M. and Sharma P. [155,156] investigated a metal hydride reactor containing about 50 kg La_0.9_Ce_0.1_Ni_5_ using a numerical simulation method. A standard pipe size with a nominal diameter of 6 inches was chosen, which gives an internal diameter of 154.06 mm and an external diameter of 168.28 mm. Cellular honeycombs structures were used to provide a better heat transfer in the metal hydride bed. It was noted that the overall productivity of the metal hydride reactor was higher when the hexagonal honeycomb heat transfer enhancements were located at a distance of 3 cm from each other. With a further decrease in the distance between structures, the improvement in the reaction rate was insignificant. According to observations, the optimal axial distance between two consecutive elements was 3 cm. Using the simulation results, the authors also determined the thickness (0.9795 cm) and length (2.7438 cm) of the honeycomb heat transfer enhancements. The authors found that the addition of a hexagonal honeycomb heat transfer enhancements improved the absorption capacity of the hydride bed by more than 30%.

Singh A. et al. [133,157] suggested using copper disks with a large number of holes to improve heat transfer in the metal hydride bed (Figure 18). Thus, four configurations with different numbers of holes and the same surface area of the fins were considered [156]. After conducting numerical simulations, the authors found that the charging time for 10 g of hydrogen was 614 s, 560 s, 582 s and 604 s for Design 1, Design 2, Design 3 and Design 4, respectively.

In their other work, the authors analyzed the influence of the number of fins and their thickness. For the reactor analyzed, an increase in the number of fins from 4 to 13 leads to a reduction in the absorption time of 12 g of hydrogen from 1180 s to 610 s. Increasing the thickness of the fins also has a positive effect on the reaction time. Singh A. et al. [133] also demonstrated that fins with a larger radius, as well as greater thickness, can significantly improve thermal conductivity (Figure 19).

The same conclusions were also reached in the works of Garrison S. L. et al. [93], Ma J. et al. [125] and Nyamsi S. N. et al. [158]. They also found a positive effect of increasing the number and thickness of transverse fins on thermal conductivity. However, the authors argue that the perforation of the fins has a negligible effect on the bed temperature and reaction time.

Another modification of lamellar fins to improve heat transfer was proposed by Chandra S. et al. [159]. The authors proposed the use of conical fins, which were used in conjunction with cooling tubes (Figure 20a).

In their work, a 5 kg LaNi_5_-based system equipped with 10, 13, and 19 fins with 2, 4, and 6 cooling tubes were investigated. Copper was used as the material for tubes and fins. The authors argued that 19 or more conical fins combined with 4 cooling tubes performed better than 10 fins with the same number of cooling tubes. Later, Ayub I. et al. [160] conducted research on the methodology for optimal design of a metal hydride reactor with conical fins. In their work, the dependences of the number of fins and their pitch on the length of the cooling tube were obtained, as well as the relationship between the parameters of conical fin angle and radius. Based on the tapered fin design, the authors considered five different configurations. The largest surface area V (Figure 20b) of the shaped heat exchanger provided a higher hydrogen sorption rate and faster temperature control. 

Prasad J. S. and Muthukumar P. [161] considered three configurations of annular metal hydride reactors. The first configuration and the second configuration differed in the heat transfer fluid’s flow direction, and the third configuration was the same as the second, but with the addition of radial fins without any specific geometry (Figure 21a).

The authors found that the second configuration has an advantage over the first configuration in both absorption and desorption rates. However, the difference between the absorption rates for the first configuration and the second was not so significant (Figure 21b). The addition of cross fins greatly improves the hydrogen sorption and desorption rate, occupying only 4.6% of the reactor volume.

Visaria M. et al. [94] provided a detailed description of fins with two opposite optimization parameters taken into account (Figure 22).

The authors consider the hydrogen absorption rate, which relates directly to the need for heat transfer, as well as the possibility of minimizing the weight of the metal hydride bed and the volume occupied by the fins. A 2D numerical model was used to simulate the absorption rate. The authors selected the optimal design of the fins according to the model, in which if the calculated reaction rate does not meet the target requirement, then either the total length of the fins increases, or their placement changes. If the target value of the hydrogen absorption rate is reached, then the design is optimized by reducing the overall length of the fin. In the end, the optimal value of the fin design was obtained (Figure 22b).

In addition to transverse fins, longitudinal fins are often considered in the scientific literature. Muthukumar P. et al. [162] tested a metal hydride reactor equipped with finned tube heat exchanger to study the effect of supply pressure, cold/hot fluid temperature and overall heat transfer coefficient on hydrogen storage capacity. Copper internal longitudinal fins without complex modification and an outer water jacket were used in the work of Gupta S. and Sharma V. K. [163] (Figure 23).

In this case, central pipe was used as the passage of hydrogen. The authors found that a decrease in fin length, thickness or number slows down the reaction kinetics due to the absorption of less heat by the fins. However, it has been observed that changing the geometry of the fins does not significantly affect the reaction rate. A parametric study on the influence of fin geometry on heat transfer behavior of the reactor results in the optimized fin dimension of 12 mm height, 2 mm thickness and 12 as the number of fins.

Bhouri M. et al. [100] also consider a reactor in which the metal hydride bed was cooled by an outer jacket, and the central tube with longitudinal fins was used for hydrogen flow. Using a numerical simulation, the authors found that longitudinal fins make it possible to achieve 41% improvement in the hydrogen charging rate after 720 s of the charging process. In addition, the influence of the number and thickness of fins was considered in the work. The authors note that the charging process mainly depends on the number of fins and the clearance between their tips and the internal wall of the metal hydride reactor, while the thickness of the fins has a marginal effect on the hydrogen loading rate. Nevertheless, both the number and thickness of the fins must be optimized taking into account the compromise between the resulting hydrogen loading speed enhancement and the increase in the weight of the storage system due to the additional volume and mass of the fins.

Parida A. and Muthukumar P. [164] developed a numerical model to compare the performance of three different fin configurations (longitudinal, transverse and spiral fins). The authors found that the longitudinal fins provide better thermal enhancement in the initial period of the absorption half-cycle. However, transverse fins showed better results in the case of desorption. However, over time, all fin configurations have shown similar performance under different operating temperatures (Figure 24).

Corgnale C. et al. [165] conducted an experimental study using MOF-5 as a metal hydride based on a longitudinal honeycomb aluminum structure, previously proposed by Bhouri et al. [97] (Figure 25a).

The authors noted that the honeycomb cell structure showed great potential in increasing the rate of hydrogen desorption. In addition, such a honeycomb structure has the advantages of simple manufacturing and a high volumetric capacity. A similar hexagonal honeycomb design has been proposed by George M. and Mohan G. (Figure 25b) [166]. Each hexagonal cell has a tube for coolant flow, from which fins also extend to the cell walls. Therefore, each cell of ribbed honeycombs can be considered as an assembly of T-shaped fins. The authors have optimized the geometric dimensions of the T-shaped elements in order to reduce the weight of the metal hydride reactor.

Zhang S. et al. [167] proposed several longitudinal fin designs for a straight tube heat exchanger. Configurations with straight fins, fan-shaped fins and quadratic curve-shaped fins were considered (Figure 26A–C).

The structure with quadratic curve-shaped fins was better at the beginning of the reaction and worse at the end of the reaction than structures with uniform-width longitudinal fins. Fan-shaped fins showed the worst performance overall. Thus, the authors concluded that quadratic curve-shaped fins could be useful in application, where a quick-start hydrogen storage system is needed. The authors also confirmed that an increase in the number of fins or a mass of material with high thermal conductivity enhances the heat transfer process, and therefore, configurations with different arrangements of the fins were also considered. It was determined that the staggered arrangement of a decreasing-width longitudinal fins shows superior performance in comparison with a design with an aligned fin arrangement. However, a staggered arrangement leads to worse heat transfer when applied to straight or fan-shaped width fins (Figure 26D–F). Thus, it was concluded that it is more effective to carry the heat between two longitudinal fins than to shorten the average distance of heat transfer along fins. The authors concluded that the use of an increasing number of straight fins would be beneficial both to improve heat transfer and to simplify practical manufacture, and when considering multilayer structures, the minimum number of fins should be located in the middle layer.

In recent times, attention has also been turned to nature-inspired designs. Thus, Bai X. S. et al. [168] proposed a tree-shaped longitudinal fin (Figure 27a).

A genetic algorithm was selected to optimize the structural parameters of this kind of structure (branch length, width and angle). The authors compared an optimized tree-shaped fin heat exchanger with non-optimized tree fin and radial fin heat exchangers. The tree-shaped fins heat exchanger showed a better absorption rate compared to the radial fin reactor (Figure 27b). In the simulation of the optimization of the fin distribution of the tree heat exchanger, an improvement in the sorption/desorption rate of 20% of the processes was shown compared to a conventional heat exchanger with longitudinal fins. Another heat exchanger design inspired by biological principles has been proposed by Krishna K. V. et al. [169] (Figure 27c). The authors consider two designs with different paths for the movement of the coolant and compared them with a single-tube heat exchanger with longitudinal fins. The authors found that for the longitudinally finned heat exchanger, design 1 and design 2, a storage capacity of 90% was achieved in 210, 145 and 80 s. The authors also optimized the angle of inclination of the fins. The angled keel configuration shows the best performance and is further optimized by varying the inclination angle from 3 to 9° and the number of keels from 2 to 4. The optimized 7° inclination angle design with four keels required 57 s to reach 90% storage capacity and reduced the absorption time by 73% compared to a longitudinally finned heat exchanger.

Keshari V. and Maiya M. P. [170] consider a metal hydride hydrogen storage system integrated with copper pin fins and cooling tubes as a heat exchanger, shown in Figure 28.

Compared with traditional fins, this kind of pin fin has a smaller footprint and better flexibility. The authors considered the influence of the number of fins, as well as two options for the location of the fins relative to the cooling tubes. Using numerical simulation methods, it was determined that the second design (center-to-center fixation) has better heat and mass transfer efficiency compared to the first design (side-by-side fixation).

A Table 2 was created to summarize the data presented.

Analyzing this table, we can conclude that the addition of disks has an extremely positive effect on the reaction rate and temperature of the metal hydride filling. Obviously, the increase in fin number and fin thickness leads to a greater acceleration of the hydrogen absorption process due to the increased fin surface area and decreased conduction distance from metal hydride to the fins. However, since an increase in the number of fins and their thickness leads to a decrease in the volume of the metal hydride bed, the researchers propose optimal designs for specific metal hydride reactors. In real application, the thickness of the fins needs to be well designed for a specific reactor with its geometric dimensions. A promising configuration of fins is that proposed by Visaria M. et al. [94], with geometry-optimized transverse fins, by Chandra et al. [159], with conical fins, by Keshari V. and Maiya M. P., with complex pin fins that provide more space for metal hydride powder, and by Bai X. S. et al. [168] and Krishna K. V. et al. [169], with well-optimized bio-inspired designs. Longitudinal fins are a promising solution as well. However, their location and shape must also be taken into account. Straight fins have been shown to be advantageous in terms of ease of manufacture and efficiency over fins with a decreasing thickness at the base or end of the fin [167].

### 5.3. Other Heat Exchanger Designs

Raju M. and Kumar S. [171] proposed a metal hydride cylindrical-shaped reactor configuration with multiple cooling tubes connected by aluminum fins to reduce the thermal resistance. The authors conducted a study of such parameters as supply pressure, initial temperature of the bed and coolant flow temperature. Later, in their other work [172], they considered the same reactor in terms of optimizing the geometric parameters of the heat exchanger, such as the thickness of aluminum fins, bed diameter and cooling tube diameter. For the metal hydride reactor, the fin thickness of 2 mm turned out to be the most suitable. However, such parameters as the number of tubes, geometry of fins and additional geometrical parameters were not considered in the work. It is worth noting that the authors also considered the optimization of the helical tube and shell-and-tube heat exchanger geometry. The shell-and-tube designs showed worse gravimetric and volumetric densities than the helical coil heat exchanger as a result of the simulation. This is due to the effect of turbulence of the coolant flow in the helical coil heat exchanger.

Bai X. S. et al. [173] proposed the rectangular cooling channel to optimize the metal hydride bed configuration. The authors compare the proposed heat exchanger with the longitudinal finned tube reactor and the multilayer finned tube reactor. It was observed that the rectangle heat exchange channel reactor exhibits a lower temperature and faster absorption rate as compared to the other reactors, due to more efficient thermal transportation between the metal hydride and the cooling medium. The time required to achieve 90% hydrogen storage was reduced by 40% and 38% compared to reactors equipped with longitudinal finned single-tube and multilayer finned single-tube heat exchangers, respectively (Figure 29).

Changing the tilt angle of the heat transfer channels to a smaller side has no effect on the sorption rate, but it may favorably affect the heat transfer in the central part of the heat transfer channel. The authors also noted that the addition of metal foam in heat exchange channels can decrease the absorption time by 16% compared to the rectangle heat exchange channels without metal foam embedded.

Mudawar I. [174] proposed a heat exchanger consisting of a number of identical heat exchange modules to provide better heat transfer efficiency in a metal hydride bed. Each module consisted of an aluminum base and cover plates brazed together with serpentine microchannels for coolant flowing through the module. It was noted that the effectiveness of this design is derived due to the high heat transfer coefficients provided by coolant flowing through the serpentine microchannels.

Lototskyy M. V. et al. [48] proposed a metal hydride compressor with fins formed by steel wire wound and soldered onto the hydride tube, allowing external cooling by convection (Figure 30). Another solution involves placing that “spring” inside a metal hydride reactor.

Lewis S. D. and Chippar P. [175] proposed a metal hydride reactor equipped with an embossed plate heat exchanger and studied its performance using a mathematical model. The designs of the heat exchanger with different paths of coolant flow were investigated. Parallel-type, pin-type, as well as serpentine type heat exchangers were used (Figure 31).

The authors found that the serpentine type provides better heat transfer and contributed to a faster hydrogen absorption reaction. Moreover, this heat exchanger was able to provide a more uniform temperature distribution in the metal hydride bed compared to parallel- and pin-type heat exchangers.

Some work has proposed annular reactors with a combined heating/cooling system, which may include straight cooling tubes, heating rods, an outer jacket and fins [176,177,178,179,180] (Figure 32).

In such reactors, both the cooling tubes and fins can be the subject of optimization. In addition, one of the key parameters that significantly affect the thermal properties is the thickness of the inner/outer metal hydride layers.

For large-scale hydrogen storage in hydride-forming metals, multichannel and tube bundle reactors can be used [46,125,136,181,182,183,184,185,186]. Most tube bundle reactors are similar in design to shell-and-tube heat exchangers (Figure 33).

In tube bundle reactors, the tubes are filled with hydrogen storage material, and the coolant flows outside the tubes but inside the shell. In this case, the tubes are connected by baffles, directing the fluid flow through the shell, which increases the flow path and provides the efficient removal/supply of heat to the tubes filled with metal hydride.

## 6. Additive Manufacturing for Heat Exchangers

Since the production of complex configurations of heat exchangers for metal hydride reactors is limited by traditional manufacturing methods, the use of additive manufacturing is proposed to reduce economic costs and facilitate the implementation of complex designs. Powder bed fusion, directed energy deposition, material jetting and sheet lamination can be used in order to produce the metal parts of heat exchangers [187,188,189,190,191,192,193,194,195,196,197,198,199,200,201,202,203,204,205,206,207,208,209] (Figure 34).

Metal foams used in metal hydride reactors to increase thermal conductivity have a disordered structure. Additive manufacturing technologies make it possible to obtain ordered metal matrices, the parameters of which can be optimized for the considered metal hydride reactor. Yan H. et al. manufactured a lightweight metal sandwich panel with lattice structure for thermal management systems (Figure 35a) [210].

Lattice structures were made of AISI 304 stainless steel [210]. Complex structures for heat exchangers consisting of several layers that can be produced by additive manufacture was reviewed by Kaur I. and Singh P. [212]. Figure 35b,c shows the design used the shape of a truncated octahedron in a repeating pattern to form a lattice structure. Octahedrons were made of aluminum using EBM additive manufacture and had a low mass and high porosity. Fabricating such a structure would be difficult using traditional production methods.

Complex heat exchanger designs, such as those proposed by Keshari V. and Maiya M. P. [213], can be produced using 3D printing technology with high efficiency and accuracy (Figure 36). Moreover, straight and helical copper tubes can also be manufactured via 3D printer [214].

Fratalocchi L. et al. [215] proposed highly conductive periodic open cellular structure, manufactured on a 3D printer by SLM (Figure 37).

The material for the heat exchanger was AlSi_7_Mg_0.6_. The authors proposed to use this structure to improve heat transfer in the Fischer–Tropsch reactor. It was found that the heat transfer of the conductive periodic open cellular structure reactor was superior to both packed and metal foam packed reactors, providing smaller radial temperature gradients in the catalytic bed and minor temperature drops on the walls of the reactor with high volumetric energy releases.

Additive manufacturing technologies will also be beneficial in designing and building complex transverse and longitudinal fins with a high contact area. Recently, attention has been paid to nature-inspired designs, as is the case with heat exchanger designs for metal hydride reactors. Figure 38 represents three fractal-tree-like heat exchangers, proposed by Wang G. et al. [216]. Authors using 3D printing technology to manufacture heat exchangers made of photosensitive resin.

Figure 39 shows parametric-optimized and topology-optimized designs of the heat sinks. To manufacture these heat sinks, an SLM method of manufacturing can be applied [217].

Despite the fact that many of the 3D-printed heat exchangers are designed for air cooling, it is assumed that the numerical simulation and additive manufacturing of extremely complex fins and matrices for metal hydride reactors could be a promising method for solving problems of thermal conductivity in a metal hydride bed.

## 7. Summary

An important task in the development of hydrogen storage and compression systems is to solve the problems of heat exchange in the metal hydride bed. Numerous studies have focused on improving the performance of metal hydride reactors by applying metal hydride bed modification, varying reactor configurations and changing heat exchanger designs. Based on an analysis of the scientific literature, several conclusions can be drawn. Thus, spherical-shaped reactors show slightly better performance as compared to cylindrical-shaped reactor. However, the difference is negligible, and both cylinders can be used for hydrogen storage application. Cylindrical shape overall is more practical and convenient for practical applications, which leads to a greater prevalence of cylindrical-shaped metal hydride reactor in the reviewed literature. According to the results presented in the literature, the tubular type of the metal hydride reactor appears to be more efficient in terms of gravimetric density as compared to the chamber type reactor. The porous filter provides a uniform distribution of hydrogen in the individual solid layers of the metal hydride charge. Therefore, the tubular reactor has a higher hydrogen sorption/desorption reaction rate overall. However, the volume occupied by the porous hydrogen artery needs to be taken into account when designing some heat exchangers.

Metal foams can improve heat transfer in metal hydride bed. However, the addition of metal foam into metal hydride bed can hinder the hydrogen diffusion, reduce capacity and weaken mass transfer. The addition of foam metals appears to be a good solution for small laboratory hydrogen storage systems and mobile applications, while it is less suitable for medium- and large-scale hydrogen storage. Foam metals can be used with fins or cooling tubes, greatly improving the thermal performance of the metal hydride bed. Metal hydride–ENG compacts is an alternative method to improve the heat transfer properties of a metal hydride bed. The thermal conductivity increases almost linearly with ENG content. However, both compression and ENG incorporation reduce the hydrogen permeability. Thus, the ENG content and compression pressure needs to be optimized. The most commonly used metal hydride to ENG ratios is 90:10. The number of cooling tubes must be chosen taking into account the geometric dimensions of the metal hydride reactor. Increasing the number of tubes leads to a positive effect on the heat transfer in the metal hydride bed and on the reaction rate due to the high contact area. However, further increases in the number of tubes does not lead to a significant improvement. The best arrangement and diameter of the tubes is considered to be where the sum of the distances from each point of the vessel to the cooling boundary has reached a minimum. Helical tubes almost always perform better than straight tubes. The diameter of the spiral and tube appear to be the main factors affecting heat transfer in the metal hydride bed. Integrations of concentric annular ring-shaped heat exchangers are also a good choice for enhancement cooling/heating performances with respect to standard solutions. The system was found to achieve the best performance for a symmetrical disposition of the annular ring which divides the total domain into two equal domains, halving the same bed thickness. In general, the larger the contact area between the fins and the metal hydride bed, the better the heat transfer effect. Therefore, increasing the number of fins, their thickness and diameter in the case of transverse fins leads to a better performance of the metal hydride reactor. However, longer fins have a larger surface area and therefore a higher heat transfer rate. The temperature gradient in longer fins is much larger, and, as a consequence, the efficiency drops. This effect is worse in fins with lower thermal conductivity. Considering the geometry of longitudinal fins, straight fins and fins with decreasing width show better performance. A complex design of transverse fins can be applied to reduce the weight of the heat exchanger, increase the payload in the form of metal hydride and increase the contact area between the fins and the metal hydride bed. Thus, the application of additive technologies is a promising direction for the production of heat exchangers with a complex design.

## Figures and Tables

**Figure 1 materials-16-04891-f001:**
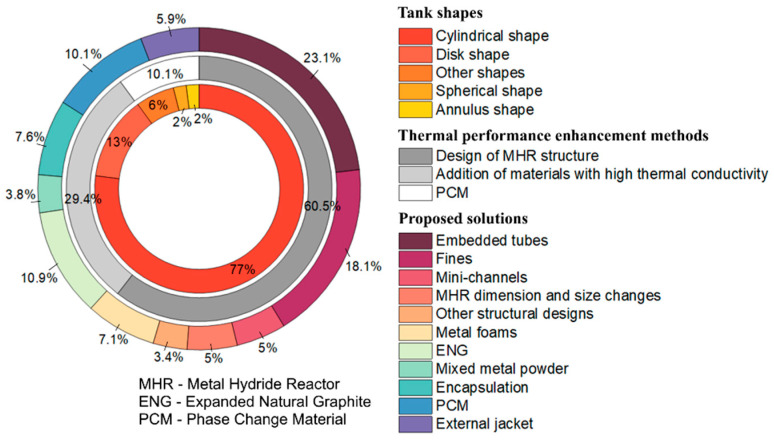
Metal hydride reactor shapes, thermal performance enhancement methods and proposed solutions, considered in scientific literature.

**Figure 2 materials-16-04891-f002:**
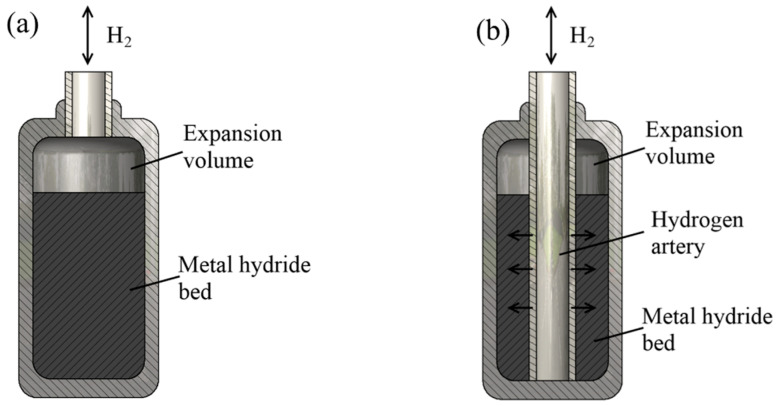
Different metal hydride reactor types: (**a**) chamber (tank) reactor; (**b**) tubular reactor.

**Figure 3 materials-16-04891-f003:**
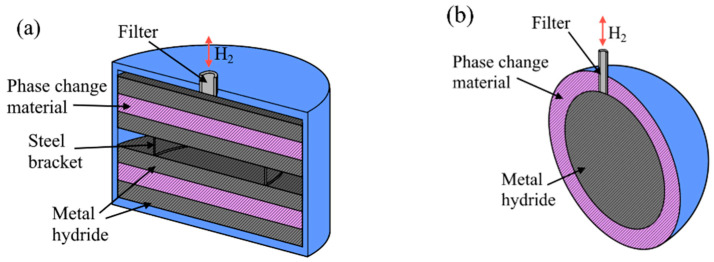
Different reactor shapes: (**a**) disk-shaped reactor and (**b**) spherical-shaped reactor.

**Figure 4 materials-16-04891-f004:**
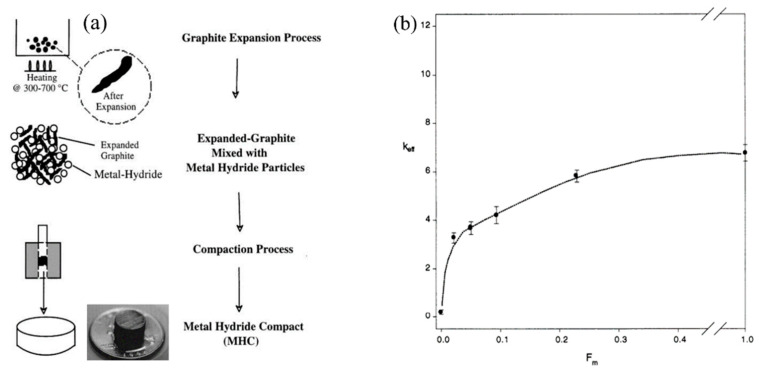
A schematic of recompressed expanded graphite technique (**a**) and measured effective thermal conductivity of metal hydride compacts versus mass fraction of graphite (**b**) [64].

**Figure 5 materials-16-04891-f005:**
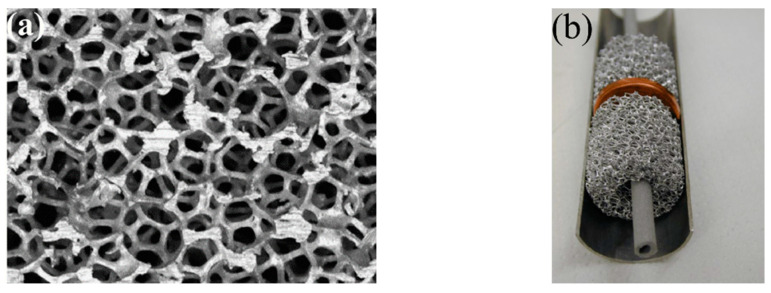
Aluminum foam (**a**) and metal hydride reactor with aluminum foam (**b**) [76].

**Figure 6 materials-16-04891-f006:**
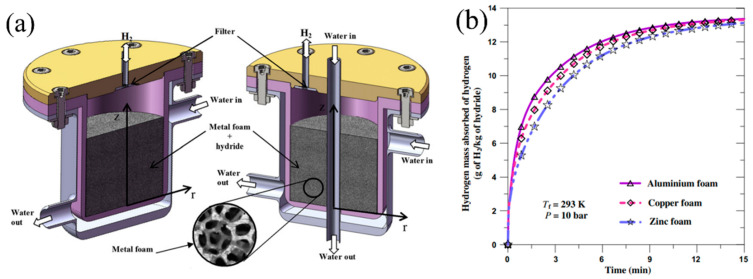
Metal hydride reactor with metal foam (**a**) and effect of base material on the storage time (**b**) [84].

**Figure 7 materials-16-04891-f007:**
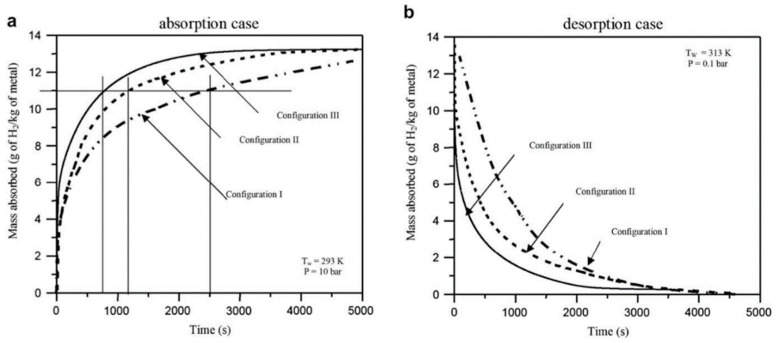
Time evolution of the hydrogen mass stored (**a**) and desorbed (**b**) in three different metal hydride reactor configurations [113].

**Figure 8 materials-16-04891-f008:**
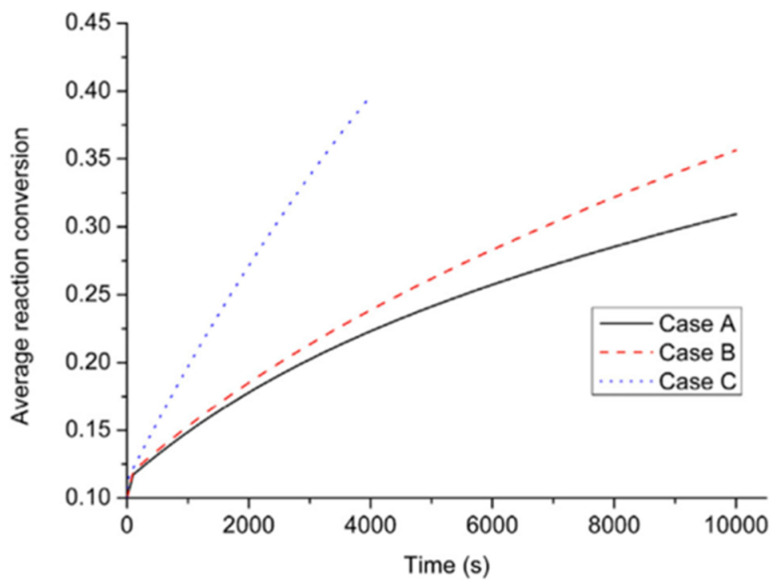
Average reaction conversion between the three reactors [117].

**Figure 9 materials-16-04891-f009:**
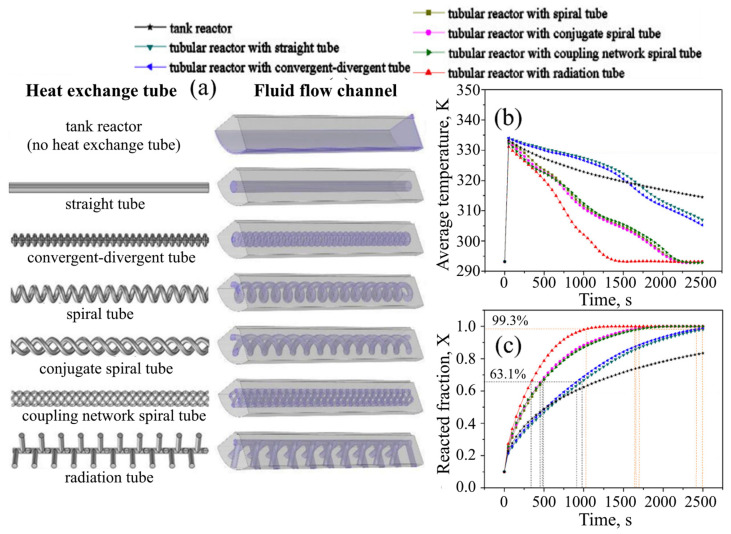
Various heat exchanger designs (**a**) and influence of design on average bed temperature (**b**) and concentration (**c**) [130]. Black dots at 0 s indicates initial conditions and the dotted lines indicates the required time to reach 63.2% (blue) or 99.3% (red) of the complete reacted fraction.

**Figure 10 materials-16-04891-f010:**
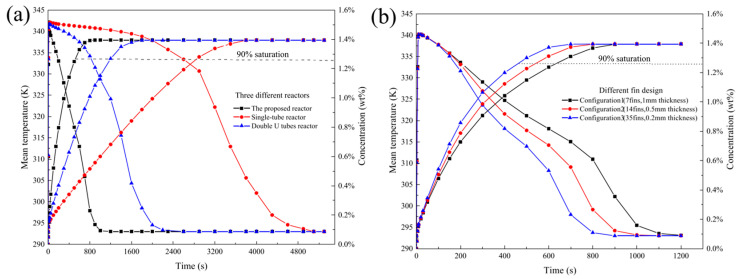
Effect of different reactors (**a**) and fin configuration (**b**) on the bed temperature and hydrogen concentration [111].

**Figure 11 materials-16-04891-f011:**
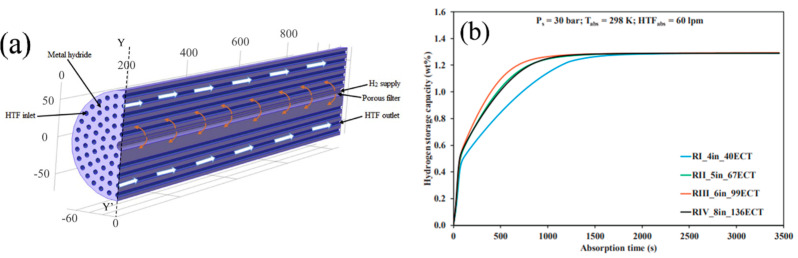
Metal hydride reactor with 99 embedded straight tubes, proposed by Raju N.N. et al. (**a**) [135], and effect of reactor geometry on variation of hydrogen storage capacity (in—diameter in inches; ECT—embedded cooling tubes; HTF—heat transfer fluid) (**b**).

**Figure 12 materials-16-04891-f012:**
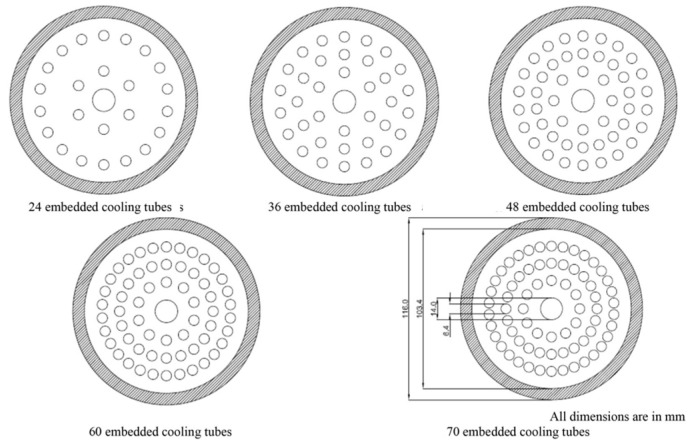
Reactor configurations with embedded cooling tubes, proposed by Anbarasu S. et al. [137].

**Figure 13 materials-16-04891-f013:**
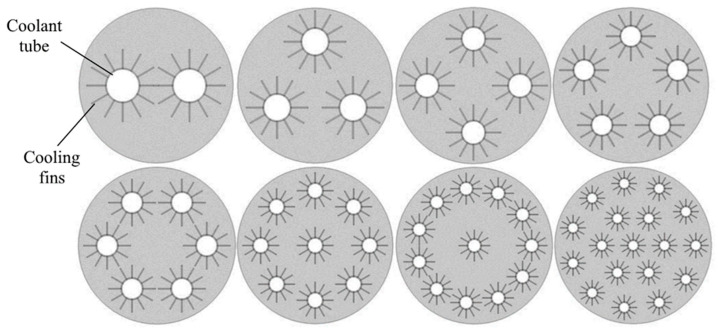
Cross-sectional configurations of the reactors with finned 2-, 3-, 4-, 5-, 6-, 9-, 12- and 18-pass heat exchangers, proposed by Keith M. D. et al. [110].

**Figure 14 materials-16-04891-f014:**
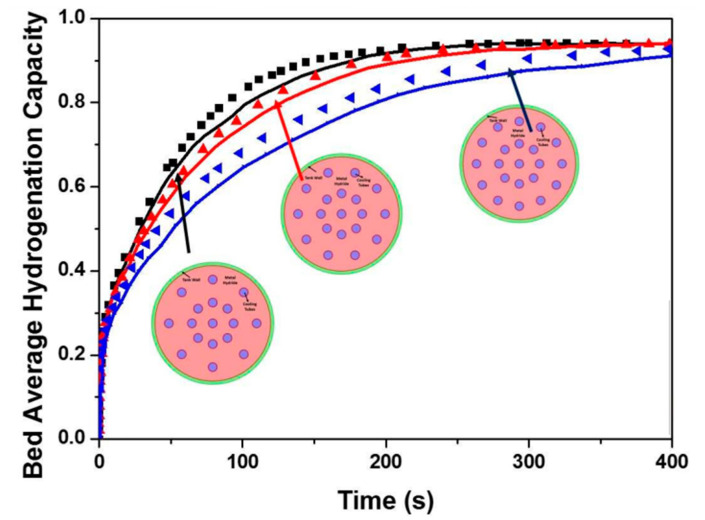
The comparison of the average hydrogenation capacity for metal hydride reactor with 12 cooling tubes (black), 16 cooling tubes (red) and cooling 20 tubes (blue). The solid line indicates the results obtained by Gkanas E. I. et al. [140], while the dotted line indicates the results obtained by Muthukumar P. et al. [139].

**Figure 15 materials-16-04891-f015:**
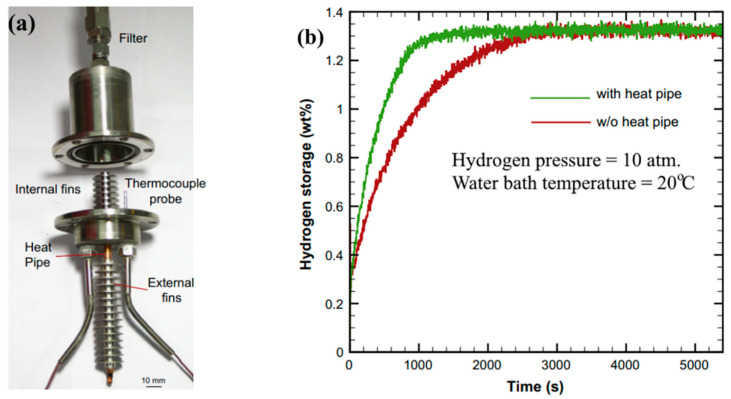
Prototype of the metal hydride reactor, proposed by Chung C. A. et al. (**a**) [151], and influence of heat exchanger on hydrogen sorption rate (**b**).

**Figure 16 materials-16-04891-f016:**
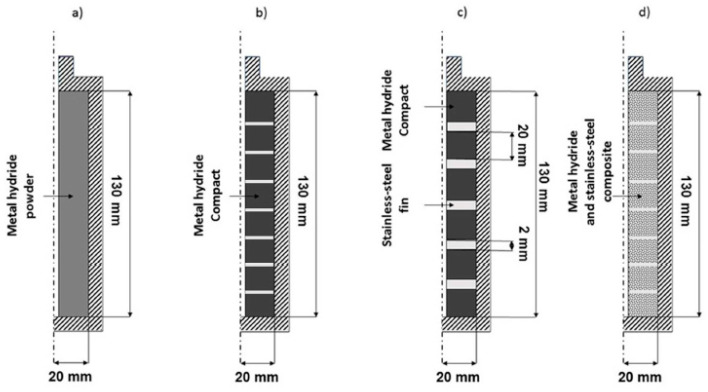
Schematic diagram of the different metal hydride bed configurations: (**a**) full Ti-Mn alloy powder bed (1st configuration), (**b**) Ti-Mn alloy compacts (2nd configuration), (**c**) alternating stainless steel fins and Ti-Mn alloy compacts (3rd configuration) and (**d**) a compact based on mixture of stainless steel and metal hydride powders (4th configuration) [152].

**Figure 17 materials-16-04891-f017:**
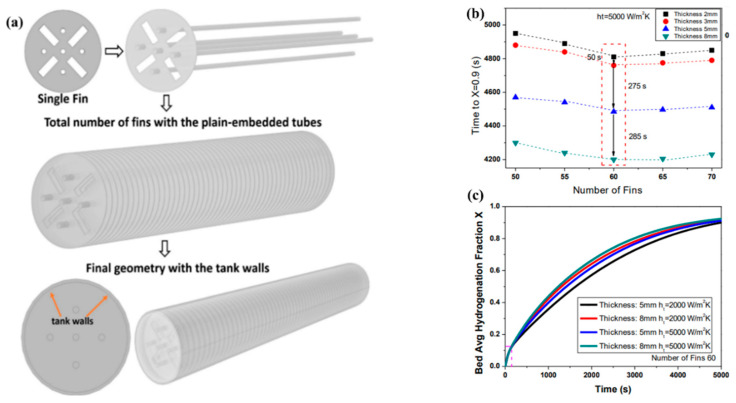
Heat exchanger design, proposed by Gkanas E. I. et al. (**a**) [154]; influence of number of fins on the time to attain 90% saturation with a dotted square area indicating the number of fins after which there is no improvement in reaction speed (**b**); and fin thickness on the average hydrogenation fraction (**c**).

**Figure 18 materials-16-04891-f018:**
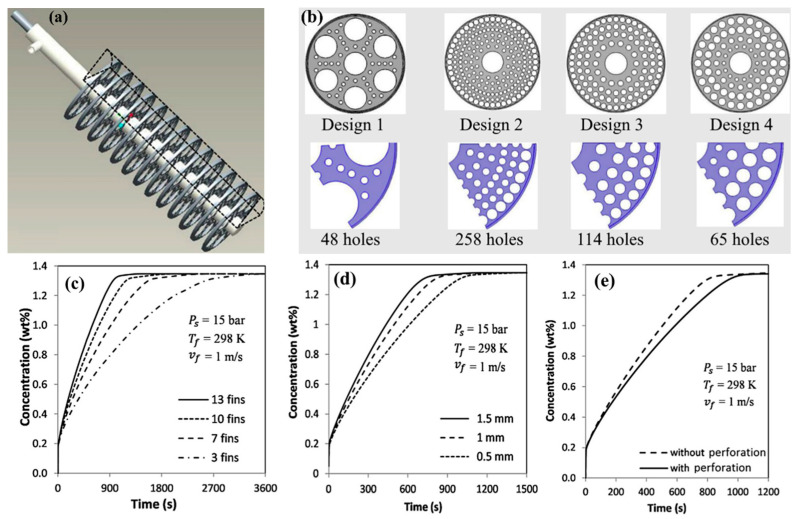
Heat exchanger (**a**), different designs of copper fins (**b**), proposed by Singh A. et al. [157], and effect of different fin number (**c**), thickness (**d**) and perforation (**e**).

**Figure 19 materials-16-04891-f019:**
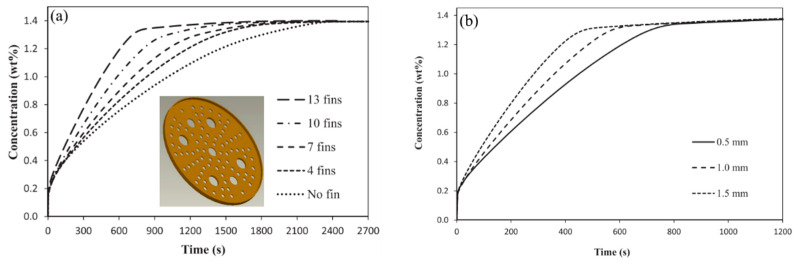
Effect of number of fins (**a**) and fin thickness (**b**) on average bed concentration [133].

**Figure 20 materials-16-04891-f020:**
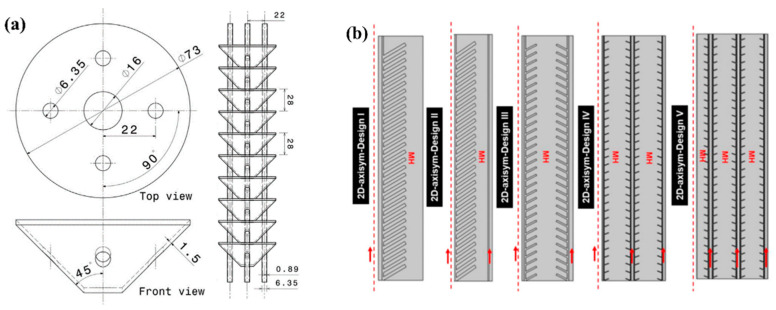
Heat exchanger with conical fins, proposed by Chandra et al. (**a**) [159] and various configuration of metal hydride reactors with conical fins, proposed by Ayub I. et al. (**b**) [160].

**Figure 21 materials-16-04891-f021:**
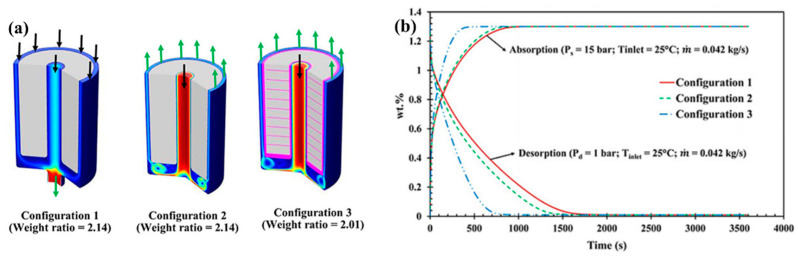
Three different configurations of annular metal hydride reactor (**a**) and hydrogen sorption/desorption rates for proposed configurations (**b**) [161]. The heat exchanger fins are highlighted in purple. The black arrow indicates the inlet of heat transfer fluid and the green arrow indicates the outlet of heat transfer fluid.

**Figure 22 materials-16-04891-f022:**
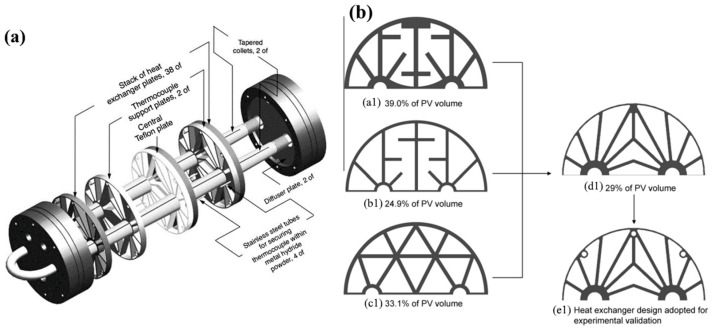
Heat exchanger (**a**) and fin geometry optimization (**b**) [94].

**Figure 23 materials-16-04891-f023:**
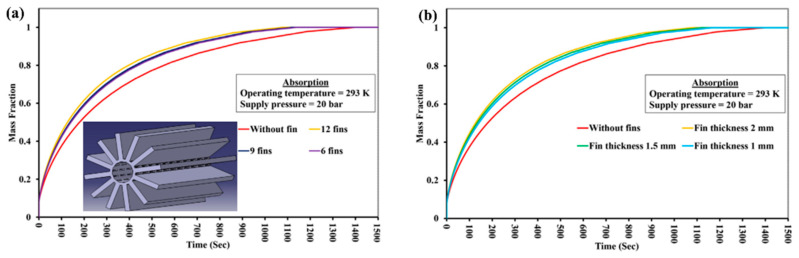
Influence of fin number (**a**) and fin thickness (**b**) on hydrogen absorption rate. Inset: finned tube heat exchanger, proposed by Gupta S. and Sharma V. K. [163].

**Figure 24 materials-16-04891-f024:**
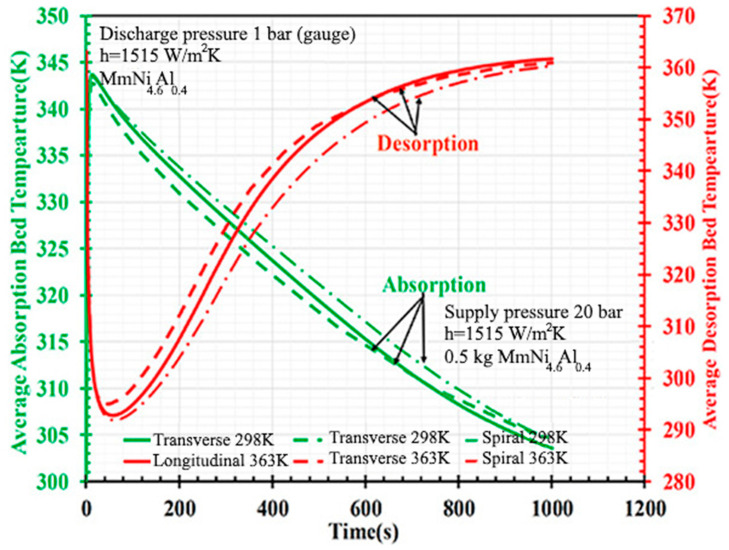
Average bed temperature for three configurations of heat exchanger, proposed by Parida A. and Muthukumar P. [164].

**Figure 25 materials-16-04891-f025:**
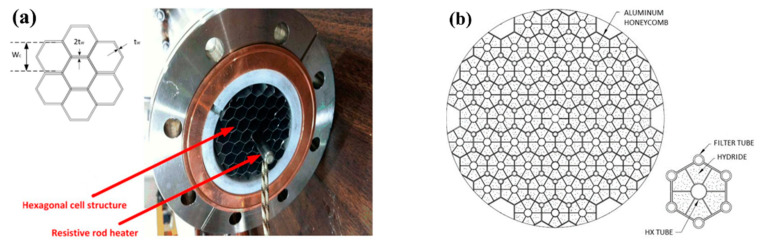
Honeycomb cell structure for metal hydride reactor, proposed by Corgnale C. et al. (**a**) [165] and George M. and Mohan G. (**b**) [166].

**Figure 26 materials-16-04891-f026:**
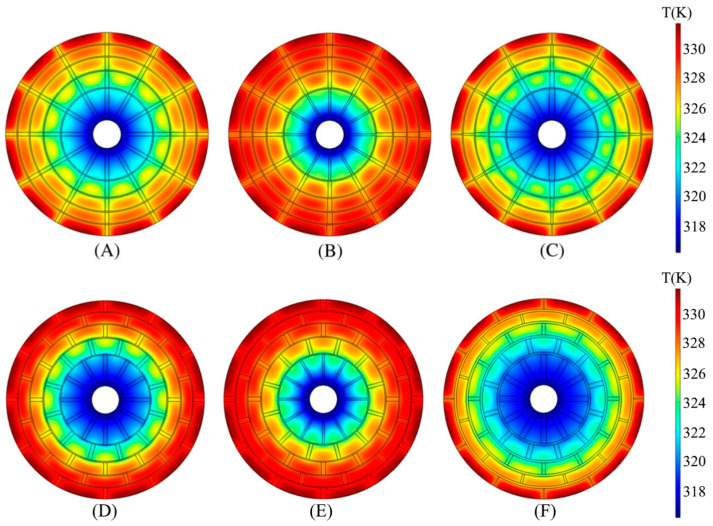
Temperature distribution of reactors with different multilayer heat exchanger configurations at 1000 s. (**A**) Aligned straight fin arrangement; (**B**) aligned fan-shaped fin arrangement; (**C**) aligned quadratic curve-shaped fin arrangement; (**D**) staggered straight fin arrangement; (**E**) staggered fan-shaped fin arrangement; (**F**) staggered quadratic curve-shaped fin arrangement [167].

**Figure 27 materials-16-04891-f027:**
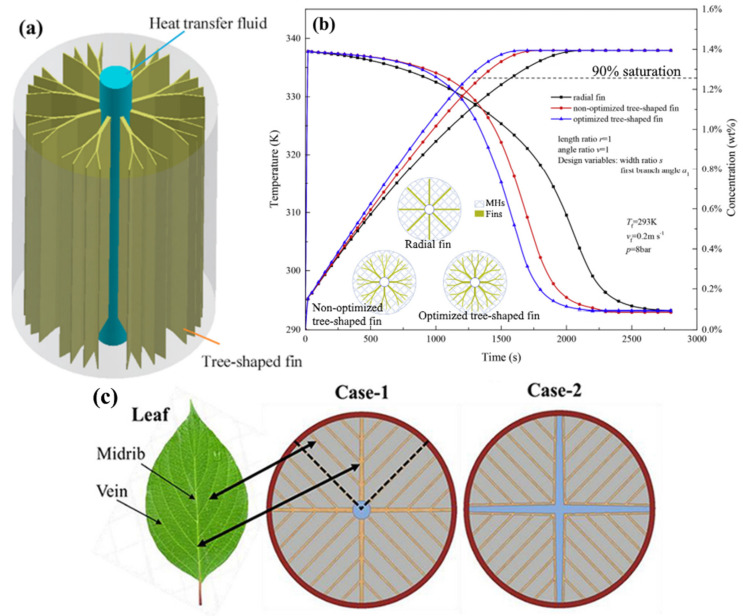
Heat exchanger with tree-shaped fins, proposed by Bai et al. (**a**) [168], comparison of the average bed temperature of tree-shaped fin heat exchanger with longitudinally finned heat exchanger (**b**) and bio-inspired heat exchangers designs, proposed by Krishna K. V. et al. (**c**) [169].

**Figure 28 materials-16-04891-f028:**
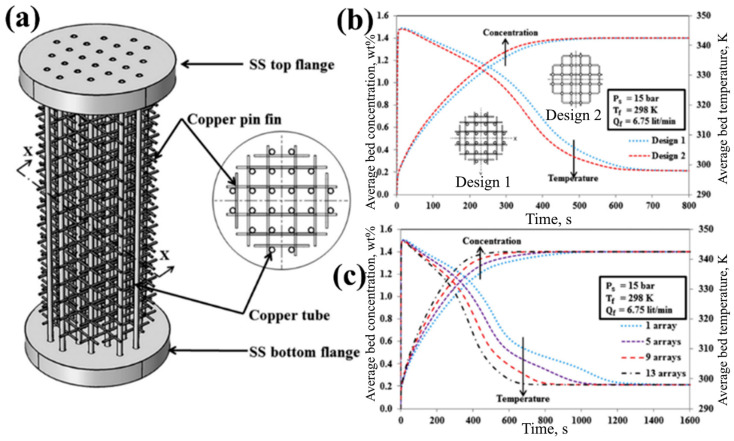
Heat exchanger with pin fins, proposed by Keshari V. and Maiya M. P. (**a**) [160], and influence of design (**b**) and number of pin fin arrays for design 1 (**c**) on average bed concentration and temperature.

**Figure 29 materials-16-04891-f029:**
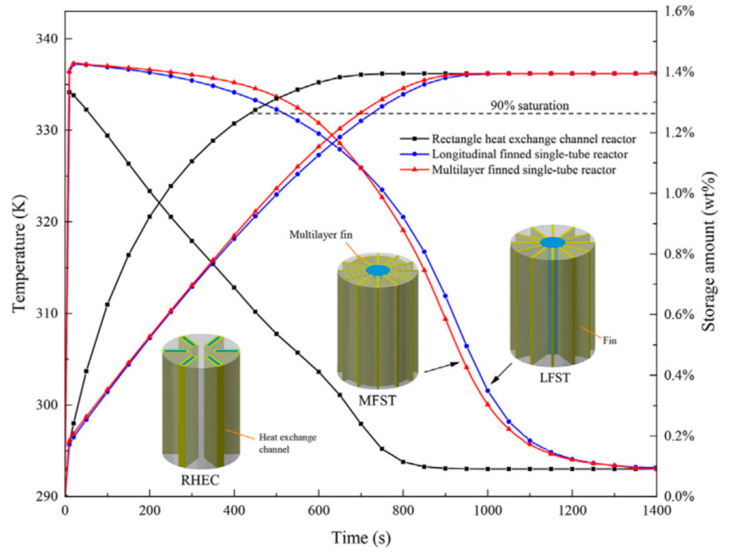
Influence of design on average bed concentration and temperature [173].

**Figure 30 materials-16-04891-f030:**
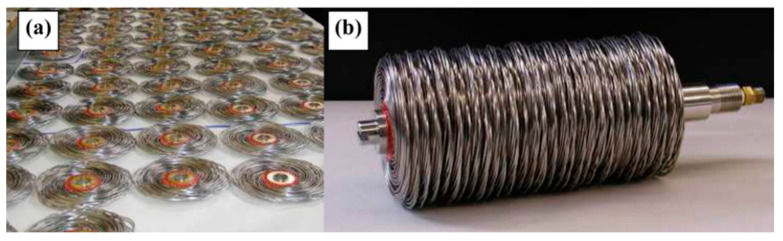
Heat exchanger for metal hydride compressor, proposed by Lototskyy M. V. (**a**); hydride tube ring manifolds; (**b**) the manifolds stacked into hydride heat exchanger [48].

**Figure 31 materials-16-04891-f031:**
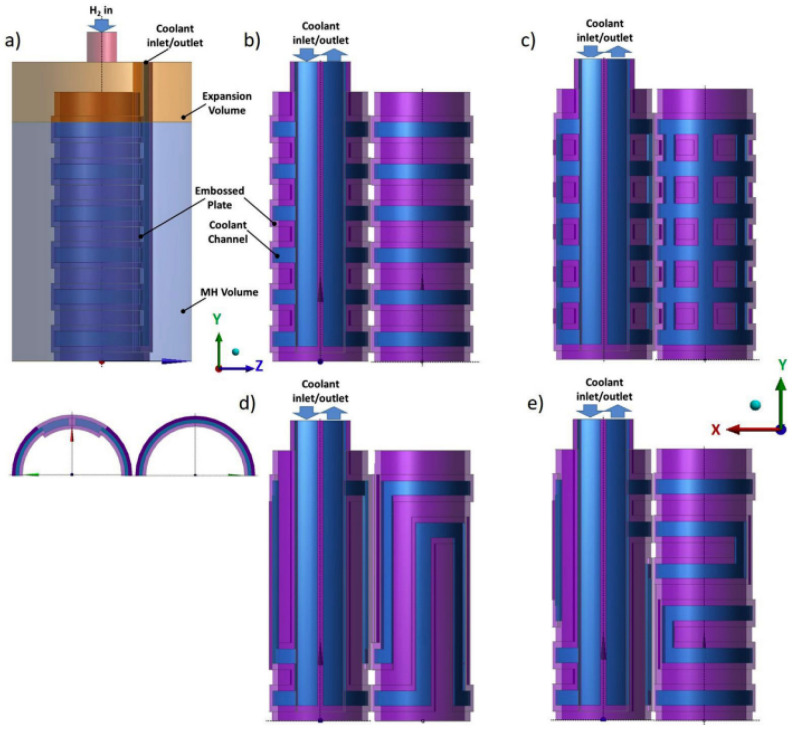
Configurations of the metal hydride storage systems with embossed plate heat exchanger, proposed by Lewis et al. [175]: (**a**) parallel flow-field embossed plate heat exchanger; (**b**) parallel-type configuration; (**c**) pin-type configuration; (**d**) vertical serpentine configuration; (**e**) horizontal serpentine configuration.

**Figure 32 materials-16-04891-f032:**
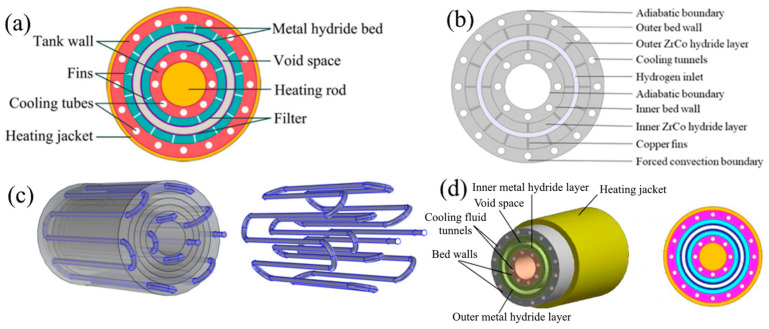
Annular reactors: (**a**) thin double-layered hydrogen ZrCo-based hydride bed, proposed by Zhang B. et al. [178]; (**b**) double-layered cylindrical ZrCo hydride bed, proposed by Li W. et al. [177]; (**c**) circular-shaped cooling pipe, proposed by Zhang B. et al. [178]; (**d**) thin double-layered annulus bed, proposed by Cui Y. et al. [176].

**Figure 33 materials-16-04891-f033:**
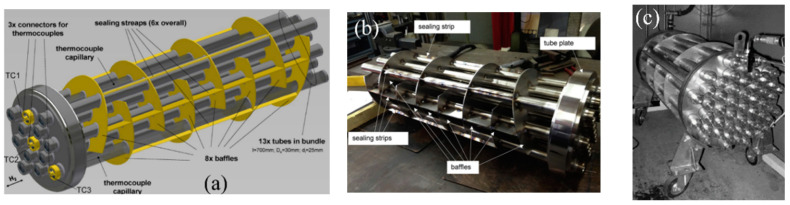
Large-scale hydrogen storage reactors: (**a**) tube bundle heat storage reactor model; (**b**) tube bundle metal hydride reactor [184]; (**c**) metal hydride storage system PX-1 [46].

**Figure 34 materials-16-04891-f034:**
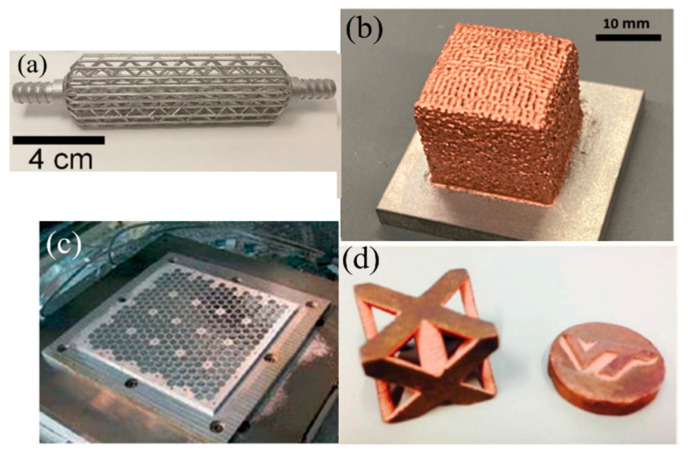
Metal parts obtained using different printing technologies: (**a**) heat exchanger, fabricated from AlSi10Mg using DMLS [207]; (**b**) 20 mm × 20 mm × 20 mm copper cube with density of 94.1% built using blue laser [200]; (**c**) aluminum honeycomb structure, manufactured by SL method [210]; (**d**) complex-shaped copper made via binder jetting [205].

**Figure 35 materials-16-04891-f035:**
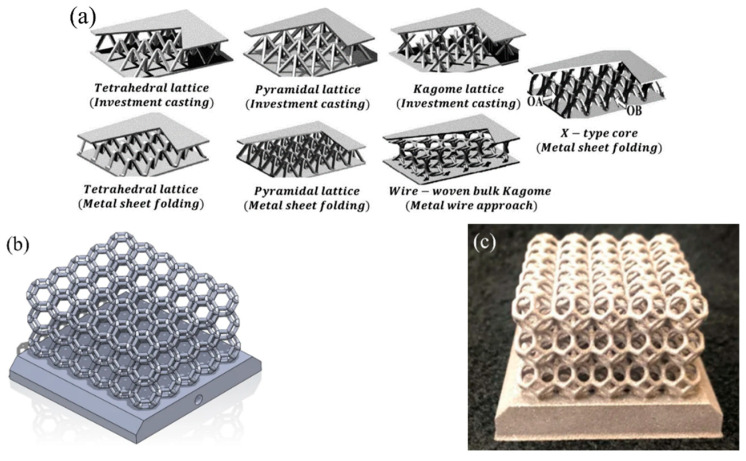
3D-printed cell structures: (**a**) metal sandwich panels with different lattice structures [210]; (**b**) lattice-structured heat sink design; and (**c**) heat sink manufactured using additive manufacturing [211].

**Figure 36 materials-16-04891-f036:**
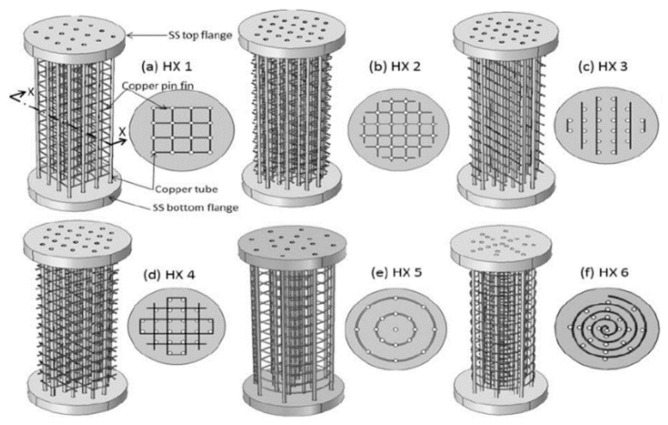
Different complex heat exchanger designs, proposed by Keshari V. and Maiya M. P. [213].

**Figure 37 materials-16-04891-f037:**
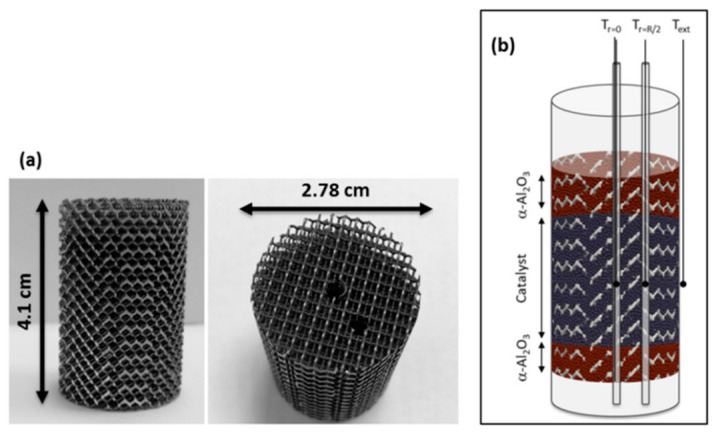
Conductive periodic open cellular structure: (**a**) manufactured heat exchanger; (**b**) qualitative scheme of the heat exchanger embedded into the Fischer–Tropsch fixed-bed reactor [215].

**Figure 38 materials-16-04891-f038:**
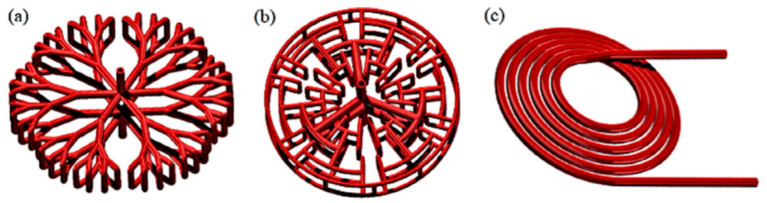
Fractal-tree-like heat sink designs: (**a**) Y-type; (**b**) H-type; and (**c**) traditional helical tube [216].

**Figure 39 materials-16-04891-f039:**
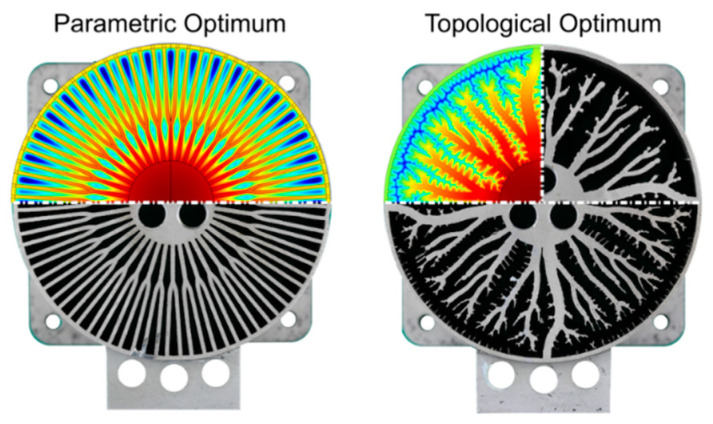
Heat sink designs: (**left**) with tree-shaped fins; (**right**) bio-inspired complex fins [217].

**Table 1 materials-16-04891-t001:** Methods for modifying metal hydride bed for hydrogen storage system.

Metal Hydride Material	Type of Modification	Heat Transfer Coefficient, W/(mK)	References
MmNi_4.15_Fe_0.85_	Aluminum foam	6.9	[81]
MmNi_4.46_Al_0.54_	Copper wire matrix	0.5–2.5	[88]
LaNi_4.85_Sn_0.15_	Expanded natural graphite flakes	19.5	[59]
LaNi_5_	Porous metallic-matrix hydride compacts with aluminum	12.3	[89]
LaNi_5_	Copper encapsulation	3.5	[90]
LaNi_4.75_Al_0.25_	Copper coating	1.78–4.3	[91]
LaNi_5_	Aluminum foam	≤10.0	[81]
LaNi_5_	Copper coating	6.0–9.0	[63]
LaNi_5_	Copper encapsulation	5.0	[58]
LaNi_5_	Copper coating	2.17–6.6	[91]
Ca_0.6_Mm_0.4_Ni_5_	Copper coating	0.8–2.8	[91]
Mg_90_Ni_10_	Pelletized hydride-graphite composites	>10.0	[92]
Hydralloy^®^ C5-based MHC	Metal hydride/ENG compacts	~10–15	[71]
La_0.8_Ce_0.2_Ni_5_	Graphite flakes	4.7	[68]
La_0.8_Ce_0.2_Ni_5_	Graphite flakes with copper wires	6.8	[68]
La_0.9_Ce_0.1_Ni_5_	Metal hydride/ENG compacts	8.1	[72]
MgH_2_	Metal hydride/ENG compacts	1.0–4.2	[93]

**Table 2 materials-16-04891-t002:** Heat exchanger designs for hydrogen storage system.

References	Hydrogen Storage Material	Heat Exchanger Design	System Characterization
Gkanas E. I. et al. [154]	MmNi_4.6_Al_0.4_	Combination of cooling tubes with fins; fins with five holes.	The optimum fin number is 60 and fin thickness 5–8 mm. The value of the heat transfer coefficient is about 2000–5000 W/(m^2^K).
Afzal M. and Sharma P. [155,156]	La_0.9_Ce_0.1_Ni_5_	Hexagonal heat transfer enhancement.	Improves the desorption rates by 20%, from 313 K to 323 K. Improves the absorption performance of the hydride bed by over 30%.
Singh A. et al. [157]	LaNi_5_	Copper fins with perforation; fin diameter—52 mm; fin thickness—0.5 mm.	The charging time for 10 g of hydrogen: 614 s (Design 1), 560 s (Design 2), 582 s (Design 3) and 604 s (Design 4).
Singh A. et al. [133]	LaNi_5_	Two “U”-shaped tubes and copper fins with perforation; fin diameter—61 mm.	The required charging time is around 610 s for a storage capacity of 12 g (1.2 wt%).
Garrison S. L. et al. [98]	A sodium alanate complex metal hydride	A transverse fin and a longitudinal fin.	Parameters for the optimal design (transverse fin): internal diameter of the cooling tube, 0.085 in.; thickness of the cooling tube, 0.020 in.; length of cooling fin, 0.290 in.; thickness of the cooling fin, 0.004 in. To store 1 kg of hydrogen would require ~97,800 unit cells with 14.9 kg of aluminum cooling tubes and cooling fins and 126 kg of hydride precursor. Parameters for the optimal design (longitudinal fin): internal diameter of the cooling tube, 0.100 in.; thickness of the cooling tube, 0.020 in.; length of cooling fin, 0.340 in.; thickness of the cooling fin, 0.004 in. To store 1 kg of hydrogen would require 537 independent cooling tubes in a 1 m-long tank vessel, with 14.7 kg of aluminum cooling tubes and cooling fins and 126 kg of hydride precursor.
Nyamsi S. N. et al. [158]	LaNi_5_	Two designs: the baseline design with a fin length of 7.8 mm and the optimized design with a fin length of 15 mm.	The hydrogen charging time is ~600 s when the optimized design is used; Increasing the cooling tube diameter can bring about 25% of hydrogen charging time.
Chandra S. et al. [159]	LaNi_5_-based system	Cylindrical reactor (OD 88.9 mm) with internal conical copper fins and cooling tubes (1/4”, SS 316); 10, 13 and 19 copper fins with 2, 4 and 6 copper cooling tubes.	Conical fins offer enhanced heat transfer. A design with 19 fins with 6 tubes requires 290 and 375 s for 80% and 90% hydrogen saturation level, respectively.
Ayub I. et al. [160]	Nano-engineered composite (MgH_2_ + V_2_O_5_)	Annular hollow truncated conical fins (steel 316L). The total length of a metal hydride reactor is 0.64 m, and the radius for all the designs is 0.12 m.	Optimal design: central heat transfer pipe + multiple jackets for MH bed and heat transfer fluid. The reaction time was 15,000 s, value of gravimetric exergy output rate was 1.23 W/kg, and value of exergy output was 0.028 kW.
Prasad J. S. and Muthukumar P. [161]	LaNi_5_	Cross fins.	Fins occupy about 4.6% of the reactor volume, improving the hydrogen sorption and desorption rate by a factor of 2.07 and 1.92.
Visaria M. et al. [94]	High-pressure metal hydride Ti_1.1_CrMn	A 260.3 mm-long prototype with aluminum plates. The heat exchanger could store 2.65 kg of metal hydride powder.	The design occupies 29% of the pressure vessel volume. The metal hydride was able to store 90% of its maximum hydrogen capacity at the end of 300 s.
Gupta S. and Sharma V. K. [163]	La_0.9_Ce_0.1_Ni_5_	Copper internal longitudinal fins without complex modification and outer water jacket. Reactor of 163 mm length and 33 mm diameter.	The optimum fin structure: 12 fins, fin height 12 mm, fin thickness 2 mm. Reduction in the overall reaction time by almost 500 s; reduction in the rise in average MH bed temperature by 22.3 K during absorption; reduction in the rise in average MH bed temperature by 6.8 K during desorption.
Bhouri M. et al. [100]	NaAlH_4_ + 2% TiCl_3_ 1/3AlCl_3_ + 0.5%FeCl_3_	Each module contains seven finned tubes spaced uniformly and arranged in a triangular array inside a cylindrical shell.	A 41% improvement in hydrogen charge rate after 720 s of charging.
Parida A. and Muthukumar P. [164]	MmNi_4.6_Al_0.4_	Three different fin configurations: longitudinal, transverse and spiral fins.	A 0.5 kg reactor can discharge hydrogen at the rate of 2.27 ppm for 2000 s.
Corgnale C. et al. [165]	MOF-5	Longitudinal honeycomb aluminum structure.	The proposed adsorption system was also shown to discharge all available hydrogen in less than 500 s; work in cryogenic conditions; and have a nominal heating power of 100 W.
George M. and Mohan G. [166]	TiCl_3_ catalyzed NaAlH_4_	Honeycomb structure.	At the optimum length/thickness ratio, the device, designed to charge 0.01 kg of hydrogen in 10 min, weighs 1.2 kg.
Zhang S. et al. [167]	LaNi_5_	Configurations with straight fins, fan-shaped fins and quadratic curve-shaped fins.	Reduction in reaction time by 25%.
Bai X. S. et al. [168]	LaNi_5_	A tree-shaped longitudinal fin.	Compared to the radial fin reactor, the hydrogen absorption time to reach 90% saturation is reduced by almost 20.7% for the optimized tree-shaped fin reactor. For the MHs thermal conductivity of 1.1 W/(mK), 3 W/(mK) and 5 W/(mK), the charging times for the mean bed temperature to reach 300 K are 1645 s, 1434 s and 1355 s, respectively.
Krishna K. V. et al. (c) [169]	LaNi_5_	Bio-inspired leaf-vein type fins.	The optimized 7° inclination angle design with four keels required 57 s to reach 90% storage capacity and reduced absorption time by 73% compared to a longitudinally finned heat exchanger.
Keshari V. and Maiya M. P. [170]	LaNi_5_	Copper pin fins and cooling tubes.	Total absorption time of 636 s with maximum storage capacity of 1.4 wt% (15 bar H_2_ gas supply pressure, heat transfer fluid temperature of 298 K, flow rate of 6.75 L/min).

## Data Availability

The raw/processed data required to reproduce these findings cannot be shared at this time as the data also forms part of an ongoing study.

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
