# Peer review of "State of the Art in Development of Heat Exchanger Geometry Optimization and Different Storage Bed Designs of a Metal Hydride Reactor"

_materials, 2023, doi:10.3390/ma16134891_

Round 1

Reviewer 1 Report

This manscript reviewed the influence of different configurations of heat exchangers and metal hydride bed for modern solid-state hydrogen storage systems. The main advantages and disadvantages of various configurations are discussed in regard to heat transfer and size characteristics.

It is a nice work, which could be accepted for publication.

Author Response

Dear Reviewer:

Please find attached for your kind review our manuscript after revision entitled “State-of-the-Art in Development of Heat Exchanger Geometry Optimization and Different Storage Bed Designs of a Metal Hydride Reactor”.

Reviewer 2 Report

This paper consists of an introduction to many metal hydride heat exchangers that have been studied and reported so far.

Trying to introduce so many different shapes and types of heat exchangers, the manuscript feels very distracting.

Unfortunately, no effort to draw Comprehensive and conclusive results such as graphs is seen in the manuscript.

Therefore, this reviewer kindly asks the authors to withdraw this manuscript and re-submit after heavy modification.

designs and methods for optimizing the geometry of heat exchangers have been actively developed as an effective way to solve heat transfer problems in metal hydride bed.

=> design methods ??

since experimental studies of the processes occurring in hydrogen storage systems are quite complex and involve high material costs

=> since experimental studies of the processes occurring in hydrogen storage systems require quite complex facilities and involve high material costs

Fig 10 (b) y axis title may be  "desorption".

It was observed, that the integration of dual helical tubes equipped with fins exhibited a significantly faster rate compared to another solutions.

=> It was observed that the integration of dual helical tubes equipped with fins exhibited a significantly faster rate compared to the other solutions.

it is required about 420 s. to fulfill the 90% storage need.

=> time unit second is 's' not 's.'

These kinds of cooling tube could also be used to design mini-channel reactor.

=> These kinds of cooling tubes could also be used to design mini-channel reactors.

Author Response

(The authors gave the same response as above.)

Reviewer 3 Report

The authors presented a proper review on development of heat exchanger geometry optimization and touched a hot topic in the field. The topic is useful and attractive. But a major problem is that the authors often list many works without proper discussion nor their own understanding. Especially when introducing the numerical simulation work in this field. Lacking of details and proper discussion. Sometimes repetitive examples are given which seem to be redundant. It’s suggested to remove some of them and keep the main body concise. As a review in the field of engineering design, it’s suggested to add some review discussions on the processing and cost parts. Some discussions on the future technical targets and outlooks of future development should be given.

Page 1 line 43:

The status of the stored hydrogen has been classified into gaseous, liquefied and absorbed state, and so do the corresponding storage methods are introduced by the author. When talking about the hydrogen that is absorbed by a medium, only the metal hydrides are discussed, however, as well-known, these hydrogen storage methods include those using physicochemical (or chemical processes). For example, in chemical process, such compounds as metal hydrides, organic or chemical hydrides, ammonia, and alloys based on magnesium and nickel, LaNi5, and silicon are used. Systems in which hydrogen storage based on physical adsorption in highly active microporous adsorbents, such as zeolites, metal–organic and covalent–organic frameworks, can also be used to fabricate energy-capacitive, explosive safe and lightweight.

Therefore, an apparent lack of introduction and discussion in hydrogen storage methods are missing here. Although, the main topic in this work is about the accessories for metal hydrides, this knowledge should be mentioned and briefly discussed.

Page 3 line 89-100.

The author shows the potential applications of additive manufacturing in heat exchanger design and manufacture optimization. But the author didn’t give out even one citation.

Subsection 5.1

The author uses coil, spiral and helical to describe the geometry of the pipes, respectively. These three words show up frequently and randomly, while indicating one same type of pipe geometry. It is recommend reducing the cumbrous verbose expression here. Choose one and make it simple.

Page 19 line 602-605.

The author highlights the outcome of a novel heat exchanger from Bai X.S, showing the giant improvements and benefits of the new design. However, no citations, source and diagrams or schematic are given. This is not appropriate and friendly to the readers. This item takes almost one paragraph, so it is worth more material, instead of pushing readers to imagine.

Page 20, line 641-646.

There is no specific geometrical data presented for the storage cylinder and the cooling tube. It’s highly recommended to give the detailed dimensions.

Page 21, line 674-682

It’s suggested to remove this part together with fig.26(a) since it has already been properly concluded before, and this part could not give new knowledge.

The authors mentioned simulation work, but no discussion on the methodology nor the conditions can be found. It’s suggested to give proper opinions on this part of work reviewed. Must be improved.

Fig. 37.

The authors presented the designs, but the performance in heat change is not known.

Line 830-831: This argument is redundant. Suggest to delete.

Line 839-842: Difficult to understand. Suggest to rewrite.

Line 1086-1121: Not relevant, suggest to remove. Please focus on the main topic of the work.

Error 1:

Page 8 line 282.

The unit of 6.8 W/m K is wrong, it should be W/m2K.

Error 2:

Abbreviation problem.

Such as HTF used in Figure 22 and 24, but there is no explanation

The language is proper and readable.

Author Response

(The authors gave the same response as above.)

Round 2

Reviewer 2 Report

I appreciate the authors effort to summarize the current thermal control technology of metal hydride reactor.

Reviewer 3 Report

the authors have addressed all the comments raised